# Auditory mismatch responses are differentially sensitive to changes in muscarinic acetylcholine versus dopamine receptor function

Lilian Aline Weber[1]*, Sara Tomiello[1], Dario Schöbi[1], Katharina V Wellstein[1], Daniel Mueller[2], Sandra Iglesias[1], Klaas Enno Stephan[1,3]

[1]Translational Neuromodeling Unit, Institute for Biomedical Engineering, University of Zurich & ETH Zurich, Zurich, Switzerland; [2]Institute for Clinical Chemistry, University Hospital Zurich, Zurich, Switzerland; [3]Max Planck Institute for Metabolism Research, Cologne, Germany

**Abstract** The auditory mismatch negativity (MMN) has been proposed as a biomarker of NMDA receptor (NMDAR) dysfunction in schizophrenia. Such dysfunction may be caused by aberrant interactions of different neuromodulators with NMDARs, which could explain clinical heterogeneity among patients. In two studies (N = 81 each), we used a double-blind placebo-controlled between-subject design to systematically test whether auditory mismatch responses under varying levels of environmental stability are sensitive to diminishing and enhancing cholinergic vs. dopaminergic function. We found a significant drug × mismatch interaction: while the muscarinic acetylcholine receptor antagonist biperiden delayed and topographically shifted mismatch responses, particularly during high stability, this effect could not be detected for amisulpride, a dopamine D2/D3 receptor antagonist. Neither galantamine nor levodopa, which elevate acetylcholine and dopamine levels, respectively, exerted significant effects on MMN. This differential MMN sensitivity to muscarinic versus dopaminergic receptor function may prove useful for developing tests that predict individual treatment responses in schizophrenia.

*For correspondence:
weber@biomed.ee.ethz.ch

**Competing interest:** The authors declare that no competing interests exist.

## Editor's evaluation

This study adds to the considerable, but often conflicting, work on how neurotransmitter systems contribute to auditory processing dysfunction. The paper details a thorough and careful analysis of an important hypothesis from the point of view of schizophrenia research: do muscarinic and dopaminergic receptors contribute to mismatch negativity effects? The answers could be useful for future treatment allocation in psychosis. The analysis was pre-registered and departures from the planned analysis were well-motivated and clearly described.

## Introduction

The auditory mismatch negativity (MMN) is an electrophysiological response to rule violations in auditory input streams (*Näätänen et al., 2001*; *Näätänen et al., 2011*). It is commonly defined as the difference between event-related potentials (ERPs) to predictable ('standard') and surprising ('deviant') auditory events, and has been interpreted as reflecting the update of a predictive (generative) model of the acoustic environment (*Winkler, 2007*; *Garrido et al., 2009*; *Lieder et al., 2013a*; *Lieder et al., 2013b*; *Weber et al., 2020*).

 

The auditory MMN is of major interest for translational research in psychiatry. First, there is strong evidence that MMN amplitudes are significantly reduced in patients with schizophrenia (for meta-analyses, see *Umbricht and Krljes, 2005*; *Erickson et al., 2016*; *Avissar et al., 2018*). Second, numerous studies in animals and humans have demonstrated convincingly that the MMN is sensitive to pharmacological alterations of NMDA receptor (NMDAR) function (*Javitt et al., 1996*; *Umbricht et al., 2000*; *Heekeren et al., 2008*; *Schmidt et al., 2012*; *Rosburg and Kreitschmann-Andermahr, 2016*) – which, in turn, plays a major role in pathophysiological theories of schizophrenia (*Olney and Farber, 1995*; *Friston, 1998*; *Goff and Coyle, 2001*; *Stephan et al., 2006*; *Stephan et al., 2009*; *Corlett et al., 2011*; *Corlett et al., 2016*; *Javitt, 2012*; *Friston et al., 2016*). The MMN has thus been suggested as a potential readout of NMDA receptor (NMDAR) hypofunction in schizophrenia and has been proposed as a promising translational biomarker (*Light and Näätänen, 2013*; *Todd et al., 2013*; *Näätänen et al., 2015*).

We have recently demonstrated that the dependency of the MMN on intact NMDAR signaling can be understood in terms of hierarchical inference about the world's statistical structure (*Weber et al., 2020*): After blocking NMDARs with ketamine, the MMN still reflected updates about the regularity in the tone sequence (lower level prediction errors, PEs), but updates about the stability of this regularity (higher level PEs) were significantly reduced. Theoretical accounts (*Behrens et al., 2007*; *Mathys et al., 2011*) predict that the level of stability in the environment should impact on lower level belief updates by scaling the certainty with which beliefs are held (the precision-weight on the PE). Consistent with this, the MMN has previously also been found to be sensitive to the overall level of volatility within a block, such that MMN amplitudes were higher in blocks with more stable regularities (*Todd et al., 2014*; *Dzafic et al., 2020*).

Here, we investigate whether auditory mismatch responses and their dependence on environmental volatility are differentially sensitive to cholinergic versus dopaminergic challenges. Acetylcholine (ACh) and dopamine (DA) are two modulatory transmitters with a general capacity to modulate NMDAR function (*Hallett et al., 2006*; *Lin et al., 2010*; *Zappettini et al., 2014*; *Zwart et al., 2018*; for review, see *Gu, 2002*), and their relative contribution to NMDAR *dys*regulation has been suggested as a major cause of heterogeneity in clinical trajectories among patients with schizophrenia ('dysconnection hypothesis', *Stephan et al., 2006*; *Stephan et al., 2009*).

Understanding the substantial heterogeneity within patient populations under the current syndromatic diagnostic categories is one of the main challenges for psychiatry and an essential basis for individualized treatment predictions (*Kapur et al., 2012*; *Stephan et al., 2017*). As a consequence, biomarkers are sought that *differentiate* between alternative pathophysiological mechanisms where, ideally, these mechanisms relate to different available treatment options.

Critically, detecting alterations of cholinergic and dopaminergic neuromodulatory transmitter systems may indeed be relevant for treatment choice in schizophrenia: while standard antipsychotic treatment options in schizophrenia rely on antagonism at D2/D3 dopaminergic receptors, they show considerable variability in their binding capacity to other receptors (*Nasrallah, 2008*). Most notably, some of the most potent antipsychotics (olanzapine and clozapine) have strong affinity to cholinergic (specifically: muscarinic) receptors (*Lavalaye et al., 2001*), in contrast to almost all other second generation antipsychotics. Therefore, a readout of the functional status of muscarinic vs. dopaminergic systems in the individual could prove valuable for understanding the neurobiological basis of differential treatment responses in schizophrenia, and, subsequently, for guiding treatment (*Stephan et al., 2009*; *Stephan et al., 2015*).

However, whether such a readout of muscarinic vs. dopaminergic function could be obtained from MMN responses is not clear. While nicotinic stimulation has been demonstrated to enhance MMN amplitudes (*Harkrider and Hedrick, 2005*; *Inami et al., 2005*; *Inami et al., 2007*; *Baldeweg et al., 2006*; *Dunbar et al., 2007*; *Martin et al., 2009*; *Fisher et al., 2012*; *Knott et al., 2012*; *Hamilton et al., 2018*), the role of muscarinic cholinergic receptors for MMN is less well established. The few human studies investigating the effects of muscarinic antagonists scopolamine and biperiden on auditory mismatch processing were inconclusive and showed mixed results (*Pekkonen et al., 2001*; *Pekkonen et al., 2005*; *Klinkenberg et al., 2013*; *Caldenhove et al., 2017*). Similarly, while several pharmacological studies of DA failed to show significant effects on MMN (*Kähkönen et al., 2002*; *Leung et al., 2007*; *Leung et al., 2010*; *Korostenskaja et al., 2008*), other studies reported significant alterations of MMN by antipsychotic drug treatment, hinting at a possible effect of DA

(*Kähkönen et al., 2001*; *Zhou et al., 2013*). However, the latter interpretation is vague, given that the antipsychotic drugs studied affect numerous types of receptors.

In summary, there is inconclusive evidence concerning the sensitivity of the auditory MMN to dopaminergic and muscarinic alterations. This could be due to small sample sizes, unspecific drugs (such as antipsychotics), and/or individual differences in pharmacokinetics and thus variability in actual drug plasma levels across participants.

Here, we report results from two double-blind, between-subject, placebo-controlled studies that address these problems and test whether auditory mismatch responses and their dependence on volatility are differentially sensitive to cholinergic and dopaminergic alterations. In study 1 (N = 81), we tested the effects of biperiden, a selective muscarinic M1 receptor antagonist, on mismatch related ERPs, and compare them to the effects of amisulpride, a selective dopaminergic D2/3 receptor antagonist. In study 2, we employed exactly the same study design, paradigm, and analysis strategy in a separate sample (N = 81), to test the impact of elevated cholinergic vs. dopaminergic transmission on mismatch amplitudes, contrasting the acetylcholinesterase inhibitor galantamine to the dopamine precursor levodopa. In both studies, we used estimates of the actual drug plasma levels at the time participants performed the experimental task in order to account for individual differences in pharmacokinetics.

Importantly, we used a new variant of an auditory oddball paradigm with explicitly varying levels of stability over time. We were interested in whether cholinergic manipulations would interact with the MMN's sensitivity to environmental volatility. This was motivated by theoretical accounts (*Mathys et al., 2011*) and experimental findings that volatility affects precision-weighting of prediction error responses, possibly via interactions of ACh and NMDA-receptor-dependent mechanisms (*Iglesias et al., 2013*; *Weber et al., 2020*). While these latter studies employed trial-by-trial computational models, our paradigm was designed to contain distinct phases of volatility. This allowed us to capture the interaction between environmental volatility and mismatch processing using the conventional approach of trial averaging and comparing mismatch responses across different phases of volatility, which maximizes sensitivity for detecting volatility effects. Here, we focus on the results of this conventional analysis, but also present the (complementary) model-based perspective on our data in Appendix 1.

Based on previous literature, one would expect mismatch responses in our paradigm to be sensitive to *(1)* volatility, with larger mismatch amplitudes during more stable phases (*Todd et al., 2014*; *Dzafic et al., 2020*; *Weber et al., 2020*), and *(2)* cholinergic manipulations, with galantamine increasing and biperiden reducing mismatch amplitudes (*Moran et al., 2013*; *Schöbi et al., 2021*). Furthermore, we expected *(3)* a differential effect of cholinergic (muscarinic) and dopaminergic receptor status on mismatch responses, as postulated by initial work on MMN-based computational assays (*Stephan et al., 2006*). Our results suggest that muscarinic receptors play a critical role for the generation of mismatch responses and their dependence on environmental volatility, whereas no such evidence was found for dopamine receptors.

## Results

We conducted two separate studies to test the effects of antagonizing dopamine and muscarinic acetylcholine receptors (study 1: amisulpride and biperiden) and of elevating dopaminergic and cholinergic signaling (study 2: levodopa and galantamine) on auditory mismatch responses.

In each of the two studies, 81 healthy male volunteers were randomly assigned to receive either placebo, or one of two study drugs, with both participants and researchers blind to drug assignment. After data exclusion, N = 71 datasets entered the group analyses for study 1 (placebo: N = 25, amisulpride: N = 24, biperiden: N = 22), and N = 78 for study 2 (N = 26 per group).

All steps of our analysis pipeline, including data exclusion criteria and statistical contrasts of interest, were specified in a time-stamped analysis plan prior to the unblinding of the researcher conducting the analysis (see Materials and Methods, section Analysis plan, data and code availability).

### MMN paradigm and distraction task

We used a new variant of the auditory oddball paradigm, in which we explicitly varied the degree of volatility in the auditory stream over time. In a classical oddball paradigm, one stimulus is less likely

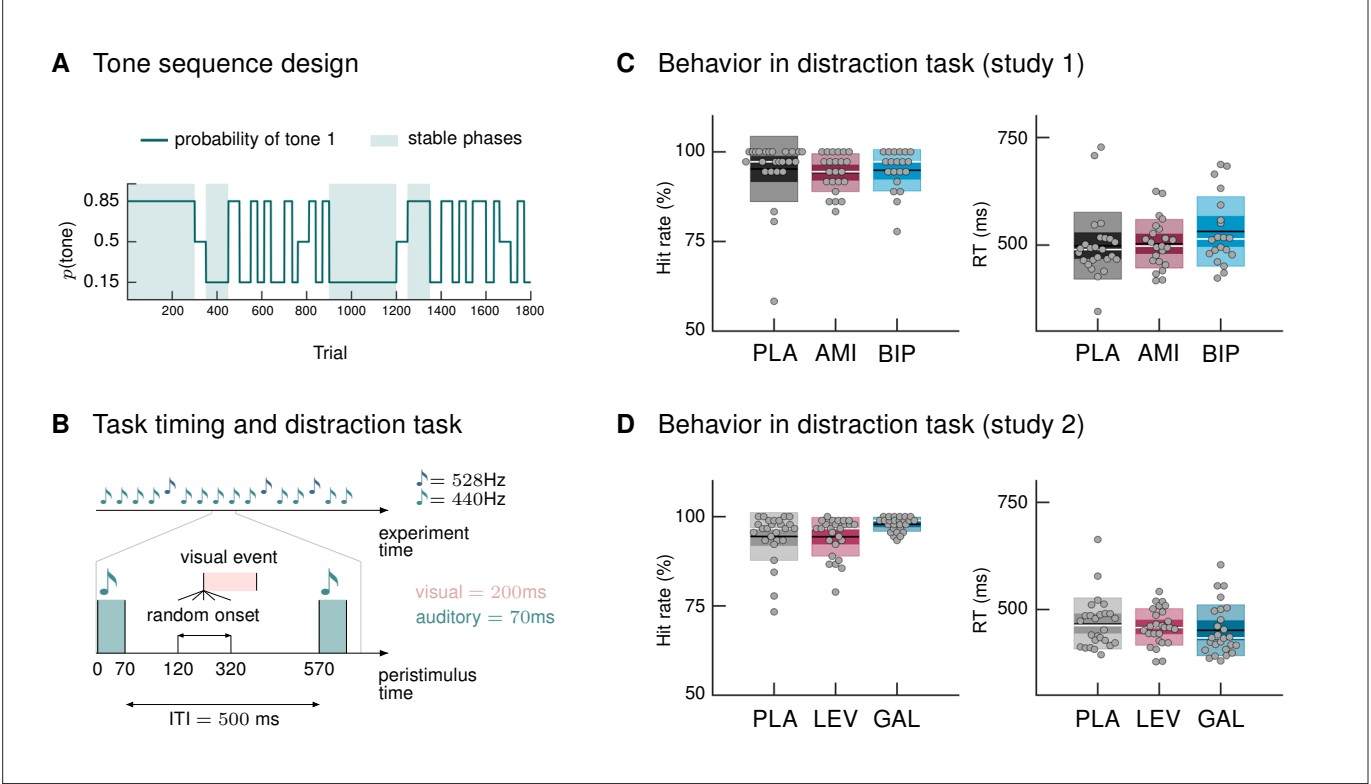

**Figure 1.** Paradigm and behavioral results. (**A**) Probability structure for the tone sequence in the oddball MMN paradigm with volatility. The probability of hearing the higher tone (tone 1, with $p$(tone 2) = 1 – $p$(tone 1)) varied over the course of the tone sequence as indicated by the blue line. Tone 1 functioned as the ‹deviant› in phases where it was less likely ($p$ = 0.15), and as the ‹standard› when it was more likely than tone 2 ($p$ = 0.85). Stable phases ($p$ constant for 100 or more trials) alternated with volatile phases ($p$ changes every 25–60 trials). (**B**) Experimental task: Overview of timing of events. Participants passively listened to a sequence of 1800 tones while performing a visual distraction task. Visual events occurred after tone presentations at a randomly varying delay between 50 and 250 ms after tone offset, in 36 (study 1) and 90 (study 2) out of 1800 trials. ITI = Inter-stimulus interval. (**C, D**) Hit rates and reaction times, per drug group, for the visual distraction task, plotted using the notBoxPlot function (***Campbell, 2020***; https://github.com/raacampbell/notBoxPlot/). Mean values are marked by black lines, medians by white lines. The dark box around the mean reflects the 95% confidence interval around the mean, and the light outer Box 1 standard deviation. (**C**) There were no significant differences between drug groups in performance on the visual distraction task. (**D**) Participants in the galantamine group had higher hit rates in the distraction task (see main text). PLA = placebo, AMI = amisulpride, BIP = biperiden, LEV = levodopa, GAL = galantamine group.

to occur and thus considered a surprising, or ‘deviant’, stimulus, whereas the other stimulus is considered the ‘standard’ event. Our sequence was generated such that both tones could be perceived as standard (predictable) or deviant (surprising), depending on the current context. Volatile phases were defined by more frequent context switches (every 25–60 trials). ***Figure 1A*** displays the probability structure underlying the tone sequence and the division into stable and volatile phases.

Following previous studies (***Garrido et al., 2008***), we only considered those tones as deviants which followed at least 5 repetitions of the other tone (resulting in $N_{deviants}$ = 119). Equivalently, we defined standards as the 6th repetition of a tone ($N_{standards}$ = 106) in order to keep trial numbers comparable across conditions. We chose this trial definition due to its specificity; however, we also considered alternative trial definitions (specified as part of our analysis plan) which allow for higher trial numbers per condition and tested the robustness of our results to this choice (we report the results of these additional analyses as figure supplements in the relevant places).

During EEG recording, participants passively listened to the tone sequence while engaging in a visual distraction task, following the suggestion that MMN assessment is optimal when the participant's attention is directed away from the auditory domain (***Näätänen, 2000***). Their task was to indicate via button press whenever they detected a centrally presented visual target (***Figure 1B***; number of targets in study 1: 36; study 2: 90). Based on the participants' responses, we calculated mean reaction times and hit rates (defined as the proportion of correct responses relative to the total number of

visual targets). Due to technical issues during measurement, behavioral data from three participants in study 1 (N = 1 amisulpride group, N = 2 biperiden group) were missing.

Participants reacted to visual targets on average after 509.8 ms (SD = 72.6; study 1) and 460.4 ms (SD = 54.2; study 2) and responded correctly to 94.8% (SD = 7.0; study 1) and 95.6% (SD = 5.3; study 2) of the presented targets. There were no significant differences between drug groups in their performance in study 1, as assessed with a one-way ANOVA for reaction times (*F* = 1.32, p = 0.27) and a Kruskal Wallis test for hit rates ($\chi^2$ = 2.92, p = 0.23, *Figure 1C*). In study 2, reaction times again did not differ significantly between drug groups (ANOVA *F* = 0.55, p = 0.58), but there was a significant effect of drug group on hit rates (Kruskal-Wallis $\chi^2$ = 8.36, p = 0.01, *Figure 1D*). Post-hoc pairwise comparisons indicated that hit rates in the galantamine group were significantly higher than in the levodopa group (p = 0.019; the difference to the placebo group failed to reach significance: p = 0.06). This result also held when excluding the participant with a hit rate below 75% (now placebo N = 25; $\chi^2$ = 8.36, p = 0.018; galantamine > levodopa p = 0.017; galantamine > placebo p = 0.104).

We cannot exclude the possibility that the participants with missing behavioral data, as well as one participant per study with very low performance level on the distraction task (hit rate << 75%, placebo groups, see *Figure 1C and D*), were paying attention to the auditory input instead of focusing on the visual task. We therefore additionally provide the results of our group ERP analysis after excluding these data sets (*Supplementary file 1*).

We analyzed trial-wise EEG responses in our paradigm using a factorial design with the within-subject factors 'mismatch' (standards versus deviants) and 'stability' (stable versus volatile; both factors implemented at the first level) and the between-subject factor 'drug' (study 1: placebo vs. amisul-pride vs. biperiden; study 2: placebo vs. levodopa vs. galantamine). To account for interindividual differences in pharmacokinetics, drug plasma concentration levels per participant (obtained via blood samples, see Materials and Methods) entered the group-level GLM as a covariate. In the following, we report the group-level effects, separately for both studies, for the main effect of mismatch, its interaction with drug group, the interaction between mismatch and stability, and the three-way interaction mismatch × stability × drug. In Appendix 2, we additionally report the main effect of stability and its interaction with drug.

## Biperiden delays and topographically shifts the MMN

In study 1, mismatch effects were different between drug groups: biperiden delayed and topographically shifted mismatch signals compared to amisulpride and placebo (see *Figure 2A* for selected sensors, and *Figure 2B* for selected time points). When considering the whole time × sensor space and correcting for multiple comparisons using Gaussian random field (GRF) theory, this difference was significant at pre-frontal sensors for the comparison between the amisulpride and the biperiden group: between 160 ms and 172 ms after tone onset, the difference between standard and deviant ERPs was significantly smaller in the biperiden group compared to the amisulpride group, peaking at 164 ms (*t* = 4.45, p = 0.012, *Figure 2C*, *Table 1*).

To understand whether this mismatch × drug interaction was driven more by standard or deviant tones or both, we compared responses to standards and deviants separately within the significant cluster (cluster 1 in *Table 1C*, k = 31). Both effects were visible: responses to standards were more positive under biperiden compared to amisulpride (Cohen's *d* = 0.83 at the peak voxel, *t* = 2.79) and responses to deviant tones were less positive under biperiden compared to amisulpride (peak *d* = 0.92, *t* = 3.15), together leading to a reduced mismatch (MMN) under biperiden.

No additional clusters showed significant mismatch × drug interactions when constraining the search volume to the significant average mismatch effect using the functionally defined mask. However, because mismatch effects in our large sample were significant in large portions of the time × sensor space (see *Figure 2—figure supplement 1*), this mask was rather unspecific. We therefore decided to deviate from our a priori analysis plan and constrain our search volume further by considering only those parts of the time × sensor space which both showed significant effects of mismatch in our sample *and* corresponded to the classical time windows and sensor locations for the mismatch negativity. In particular, we used the large cluster of frontal, fronto-central and central sensors described above which showed significant mismatch negativity between 100ms and 232 ms (peak *t* = 16.59) as a mask to constrain the search volume and subsequently constrain the multiple comparison correction to this volume using SPM's small volume correction (SVC). When focusing on this subspace, an

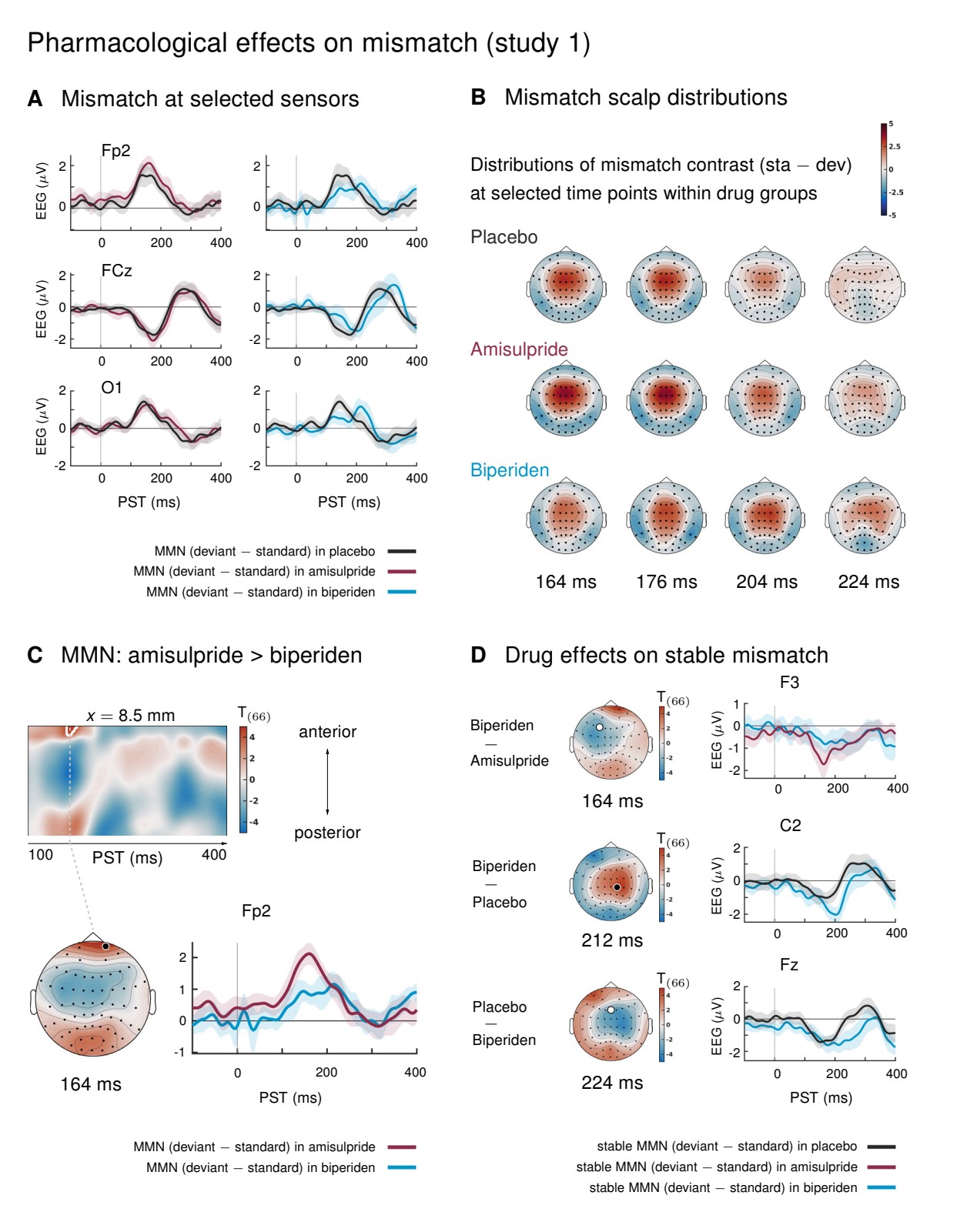

**Figure 2.** Pharmacological effects on mismatch ERPs in study 1 (interaction mismatch × drug). (**A**) Difference waves (deviants – standards) at selected sensors for the different drug groups in study 1. (**B**) Scalp distribution of the mismatch contrast at selected time points. The mismatch response in the biperiden group peaked later and more towards right central channels than in the other groups. (**C**) Mismatch responses in pre-frontal sensors were significantly weaker in the biperiden group compared to the amisulpride group. Displayed are *t*-maps for the contrast MMN amisulpride > MMN

*Figure 2 continued on next page*

*Figure 2 continued*

biperiden. The first map runs across the scalp dimension *y* (from posterior to anterior, y-axis), and across peristimulus time (x-axis), at the spatial x-location indicated above the map. Significant *t* values (p < 0.05, whole-volume FWE-corrected at the peak-level) are marked by white contours. The scalp map below shows the *t*-map at the indicated peristimulus time point, corresponding to the peak of that cluster, across a 2D representation of the sensor layout. ERP plot shows the difference waves for a selected sensor, separately for the biperiden and amisulpride groups. The location of the chosen sensor on the scalp is marked on the scalp map by the corresponding symbol. (**D**) Pharmacological effects when only testing mismatch ERPs during stable phases. Logic of display as in Panel C.

The online version of this article includes the following figure supplement(s) for figure 2:

**Figure supplement 1.** Main effect of mismatch in study 1.

**Figure supplement 2.** Main effect of mismatch in study 2.

**Figure supplement 3.** Main effect of mismatch and interaction mismatch × drug group under the new pre-processing pipeline and trial definition.

additional cluster showed a significant effect of drug on mismatch: mismatch signals were stronger in the biperiden group compared to the placebo group at right central and centro-parietal sensors with peak difference at 200 ms (*t* = 3.72, p = 0.048 after SVC). In this cluster (k = 16), only the effect on the deviant responses showed a large effect size (deviant responses were more negative under biperiden with a peak *d* = 1.13, *t* = 3.88), while the effect on standard responses was negligible (peak *d* = 0.25, *t* = 0.86), suggesting that this later interaction effect is mainly driven by a modulation of responses to deviant tones.

Together, these differences are indicative of both a delay and a shift in topography of mismatch signals in the biperiden group compared to the other two groups, leading to weaker mismatch early on, particularly in pre-frontal and frontal channels, but stronger mismatch later on, particularly in right centro-parietal channels (see *Figure 2B* for a visualization).

**Table 1.** Significant clusters of activation for main effect of mismatch (standards versus deviants) and pharmacological effects on mismatch in study 1.

The table lists the peak coordinates (*x*, *y*, and *z* for time), peak *t* values, corresponding *Z* values, whole-volume FWE-corrected *p*-values at the peak level, and cluster size ($k_E$). The last column lists the minimal and maximal time points of the cluster, i.e., the significant time window $t_{sig}$.

| Study 1: Mismatch | cluster | $x$ [mm] | $y$ [mm] | $z$ [ms] | $t_{66}$ | $Z_\equiv$ | $p_{FWE}$ | $k_E$ | $tw_{sig}$[ms] |
|---|---|---|---|---|---|---|---|---|---|
| **A** standards > deviants | 1 | −13 | 2 | 172 | 16.59 | Inf | 0.000 | 8,217 | 100–232 |
| | 2 | 0 | 13 | 400 | 8.24 | 6.81 | 0.000 | 1,037 | 364–400 |
| | 3 | −13 | 50 | 276 | 6.66 | 5.81 | 0.000 | 284 | 240–300 |
| | 4 | -4 | −95 | 304 | 5.38 | 4.88 | 0.002 | 135 | 284–328 |
| | 5 | −60 | −57 | 268 | 4.76 | 4.40 | 0.015 | 166 | 240–280 |
| | | −60 | −36 | 260 | 4.73 | 4.38 | 0.016 | | |
| | | −60 | −46 | 264 | 4.73 | 4.37 | 0.016 | | |
| **B** deviants > standards | 1 | −47 | −68 | 176 | 14.42 | Inf | 0.000 | 6,293 | 100–328 |
| | | 64 | −62 | 200 | 13.55 | Inf | 0.000 | | |
| | | 42 | −78 | 164 | 11.83 | Inf | 0.000 | | |
| | 2 | 17 | 72 | 168 | 13.93 | Inf | 0.000 | 1,592 | 100–236 |
| | 3 | 34 | −46 | 364 | 6.10 | 5.41 | 0.000 | 336 | 352–400 |
| | | 47 | −52 | 400 | 5.27 | 4.80 | 0.003 | | |
| | 4 | −51 | −30 | 400 | 5.79 | 5.19 | 0.000 | 286 | 376–400 |
| | 5 | 4 | 72 | 400 | 4.56 | 4.24 | 0.027 | 5 | 400–400 |
| | 6 | 26 | 67 | 400 | 4.46 | 4.15 | 0.036 | 1 | 400–400 |
| **C** AMI > BIP | 1 | 8 | 67 | 164 | 4.83 | 4.45 | 0.012 | 31 | 160–172 |

*Figure 2—figure supplement 1* displays the main effect of mismatch (averaging across drug groups) in our paradigm. This contrast served to confirm that our paradigm elicited a classical mismatch negativity comparable with previous reports: in a large cluster of frontal, fronto-central, and central sensors, ERPs to standard tones were significantly more positive than ERPs to deviant tones from 100 to 232 ms after tone onset, with a peak difference at 172 ms ($t = 16.59$, $p < 0.001$). The reverse was true at pre-frontal (100 to 236 ms, peak at 168 ms, $t = 13.93$, $p < 0.001$) and temporo-parietal sensors (100 to 328 ms, peak at 176 ms, $t = 14.42$, $p < 0.001$). We found eight additional clusters of significant differences between standard and deviant ERPs at later time points within peristimulus time, which are listed in *Table 1* and partly displayed in *Figure 2—figure supplement 1*.

## Neither galantamine nor levodopa affect mismatch responses

In study 2, there were no significant differences in mismatch ERPs between drug groups, both when considering the whole time × sensor space and when constraining the search volume to the significant average mismatch effect using the functionally defined mask. This also held when, by the same argument as in study 1, constraining the search even further by considering as a functional mask only the large cluster of frontal, fronto-central and central sensors described below, which corresponded to the classical time windows and sensor locations for the mismatch negativity.

Averaging across drug groups (main effect of mismatch), we again found the typical mismatch negativity effect, where ERPs to standard tones were significantly more positive than ERPs to deviant tones from 100 to 216 ms after tone onset in a large cluster of frontal, fronto-central, and central

**Table 2.** Significant clusters of activation for main effect of mismatch (standards versus deviants) in study 2.
Columns are organized as in *Table 1*.

| Study 2: Mismatch | cluster | $x$ [mm] | $y$ [mm] | $z$ [ms] | $t_{73}$ | $Z_\equiv$ | $p_{FWE}$ | $k_E$ | $tw_{sig}$[ms] |
|---|---|---|---|---|---|---|---|---|---|
| **A** standards > deviants | 1 | 13 | -9 | 176 | 14.13 | Inf | 0.000 | 7,583 | 100–216 |
| | | 4 | 18 | 160 | 13.76 | Inf | 0.000 | | |
| | | 42 | −25 | 124 | 10.85 | Inf | 0.000 | | |
| | 2 | 0 | -9 | 396 | 8.66 | 7.16 | 0.000 | 1,338 | 364–400 |
| | 3 | 4 | −95 | 292 | 7.34 | 6.33 | 0.000 | 713 | 244–332 |
| | | 26 | −89 | 280 | 6.66 | 5.87 | 0.000 | | |
| | | 47 | −62 | 252 | 5.71 | 5.18 | 0.001 | | |
| | 4 | 4 | 61 | 288 | 6.14 | 5.50 | 0.000 | 320 | 256–304 |
| | | −4 | 56 | 268 | 6.07 | 5.44 | 0.000 | | |
| | 5 | −47 | −68 | 256 | 5.93 | 5.34 | 0.000 | 364 | 232–284 |
| | | −60 | −57 | 260 | 5.65 | 5.13 | 0.001 | | |
| **B** deviants > standards | 1 | −42 | −73 | 172 | 13.97 | Inf | 0.000 | 5,637 | 100–216 |
| | | 55 | −68 | 196 | 10.87 | Inf | 0.000 | | |
| | | −42 | −73 | 124 | 10.38 | Inf | 0.000 | | |
| | 2 | -8 | −30 | 256 | 7.41 | 6.38 | 0.000 | 2,876 | 232–328 |
| | | −26 | −14 | 288 | 6.83 | 5.99 | 0.000 | | |
| | | 8 | −9 | 304 | 6.67 | 5.87 | 0.000 | | |
| | 3 | 38 | −68 | 400 | 6.65 | 5.86 | 0.000 | 302 | 372–400 |
| | 4 | 4 | 72 | 388 | 6.03 | 5.41 | 0.000 | 168 | 368–400 |
| | 5 | 68 | 18 | 192 | 5.36 | 4.91 | 0.002 | 20 | 168–204 |
| | 6 | −60 | -9 | 400 | 5.26 | 4.83 | 0.003 | 153 | 388–400 |
| | | −34 | −62 | 396 | 5.04 | 4.65 | 0.005 | | |

sensors (peak at 176 ms, $t$ = 14.13, p < 0.001), and the opposite held at left temporo-parietal and parietal sensors (100 to 216 ms, peak at 172 ms, $t$ = 13.97, p < 0.001). Standard and deviant ERPs were significantly different in nine additional clusters, which are listed in *Table 2* and partly displayed in *Figure 2—figure supplement 2*.

To rule out that some of our analysis choices (average reference, weak high-pass filter, no base-line correction, and a trial definition with comparably low trial numbers per condition) were making us insensitive to potentially more subtle drug effects in our data, we reanalyzed the data from both studies using equivalent settings as in previous studies on cholinergic modulation of MMN (*Klinkenberg et al., 2013*; *Moran et al., 2013*; *Caldenhove et al., 2017*). These settings included a stronger high-pass filter and a trial definition that resulted in higher trial numbers per condition (for details, see section Control analyses in Materials and methods).

Despite a very different pre-processing strategy and a different trial definition, the main effects of mismatch in both studies were highly similar to our original findings: a typical fronto-central MMN cluster between 100 and 244 ms, followed by a fronto-central P3a like response between 232 and 352 ms, and a late P3b-like effect between 376 and 400 ms (with all fronto-central effects being mirrored by opposite-sign effects in prefrontal and temporal sensors, as is typical for the MMN, *Figure 2—figure supplement 3A,B*).

The new analysis located the most prominent pharmacological effect in a later time window compared to the results with our original pipeline: The late positive component of the mismatch waveform showed a delayed peak under biperiden, resulting in significant differences in mismatch amplitude between the biperiden group and both the amisulpride and the placebo group around 344 ms (*Figure 2—figure supplement 3C*). Such a shift in the dominant ERP component is not surprising when using a strong high-pass filter (for a critical discussion of the effects of strong high-pass filtering, see *Tanner et al., 2015*).

Critically, and consistent with our original findings, mismatch responses under this adapted pipe-line were affected by biperiden, compared to both placebo and amisulpride, while we did not find any evidence for dopaminergic effects on mismatch responses. Just as in our main analysis, there were no significant effects of drug on mismatch responses in study 2 (galantamine and levodopa).

A further difference between our analysis approach and previous reports on muscarinic and galan-tamine effects on MMN in the literature (*Klinkenberg et al., 2013*; *Moran et al., 2013*; *Caldenhove et al., 2017*) is the use of region-of-interest (ROI) analyses. To fully account for any differences in analysis approach between our and previous studies, we therefore additionally performed a region-of-interest (ROI) analysis, focusing on exactly those sensors used in these previous studies (Fz, FCz, and Cz), and following their (peak-based) approach for extracting MMN amplitudes and latencies in every participant (*Klinkenberg et al., 2013*; *Caldenhove et al., 2017*) (for details, see section Control analyses in Materials and methods).

In line with the results obtained under our original pipeline, we found that the MMN peak latency in study 1 was increased under biperiden (mean: 181.9 ms, std: 3.4) compared to placebo (mean: 168.8 ms, std: 3.2, *Table 3*). Peak amplitudes were not significantly different between drug groups. This is consistent with a temporal shift of the mismatch response in the early (classical) MMN time window of the kind we describe above. In other words, even though the whole time × sensor space analysis under the new processing pipeline had located the dominant drug effect in a later component, we still

**Table 3.** Results of the ROI analysis.

Table lists mean (std) values of the peak amplitudes and latencies separately for each drug group in the two studies. $F$ (p) values refer to the effect of the factor drug group in a 3 × 3 ANOVA (drug × sensor). Last row lists the significant post-hoc comparisons between pairs of drug groups. Lat. = Latency.

| | Study 1 | | | | Study 2 | | | |
|---|---|---|---|---|---|---|---|---|
| | PLA | AMI | BIP | $F_{2,208}(p)$ | PLA | LEV | GAL | $F$ (p) |
| Peaks (µV) | −1.83 (0.1) | −1.95 (0.1) | −2.04 (0.1) | 1.1 (*0.33*) | −1.75 (0.1) | −2.0 (0.1) | −1.76 (0.1) | 1.73 (0.18) |
| Lat. (ms) | 168.8 (3.2) | 173.7 (3.2) | 181.9 (3.4) | **4.06 (*0.02*)** | 165.1 (2.75) | 173.2 (2.75) | 163.7 (2.75) | **3.5 (*0.03*)** |
| Post-hoc $t$ | Lat.: BIP > PLA | | p = *0.013* | | Lat.: LEV >GAL | | p = *0.038* | |

**Table 4.** Significant clusters of activation for interaction effects (mismatch × stability) on ERPs in study 1.
Columns are organized as in **Table 1**.

| Study 1: mismatch × stability | cluster | $x$ [mm] | $y$ [mm] | $z$ [ms] | $t_{66}$ | $Z_{\equiv}$ | $p_{FWE}$ | $k_E$ | $tw_{sig}$[ms] |
|---|---|---|---|---|---|---|---|---|---|
| **A** stable MMN > volatile MMN | 1 | 17 | −19 | 204 | 5.73 | 5.14 | 0.001 | 591 | 180–220 |
| | | −17 | −25 | 196 | 5.68 | 5.10 | 0.001 | | |
| **B** volatile MMN > stable MMN | 1 | 42 | −78 | 200 | 5.63 | 5.07 | 0.001 | 119 | 188–236 |
| | | 55 | −68 | 200 | 5.17 | 4.72 | 0.005 | | |
| | | 64 | −62 | 200 | 5.04 | 4.62 | 0.007 | | |
| | 2 | −60 | −57 | 208 | 5.33 | 4.85 | 0.003 | 29 | 200–220 |

found evidence for this early MMN shift when focusing on the classical MMN sensors. Note, however, that the peak-based approach of extracting ERP amplitudes and latencies employed in this ROI analysis is known to be susceptible to noise (e.g. **Clayson et al., 2013**) and that we will base our main conclusions on the whole sensor space analysis presented above.

In study 2, consistent with the whole sensor-space analysis, there was no significant difference between MMN peak amplitude or latency between the galantamine group and placebo. Peak latency differed significantly between galantamine and levodopa (with galantamine peaking earlier), but neither was significantly different from placebo (**Table 3**). Due to the above-mentioned caveats with the ROI analysis approach, and the fact that this apparent latency effect was not even visible in the average ERP traces at FCz (**Figure 2—figure supplement 3**), we did not follow up on this finding.

## Mismatch responses are stronger during stability

Our paradigm allowed us to test whether auditory mismatch signals would depend on the current level of stability in the sensory input. In study 1, we found such an interaction effect (mismatch × stability) in 3 clusters. Between 180 and 220 ms after tone onset, mismatch was significantly stronger in stable as compared to volatile phases, with a peak difference at 204 ms ($t$ = 5.13, p = 0.001) at central and centro-parietal sensors. Right parietal and left temporo-parietal sensors, which generally show the mismatch effect with the opposite sign compared to fronto-central channels, also showed stronger (negative) mismatch for stable phases than for volatile phases (right parietal cluster: 188–236 ms, peak at 200 ms, $t$ = 5.07, p = 0.001; left temporo-parietal cluster: 200–220 ms, peak at 208 ms, $t$ = 5.33, p = 0.003; see **Table 4** and **Figure 3**).

Interaction effects in central channels reflected the following pattern: responses to standard tones were more positive and responses to deviant tones more negative during stable phases than during volatile phases (**Figure 3D**). The opposite was true for interaction effects at temporo-parietal clusters. This indicates that mismatch negativity, as defined by the difference between standards and deviants, was stronger during more stable periods.

In study 2, ERPs showed no significant interaction effects between the factors mismatch and stability. In other words, mismatch responses did not differ between stable and volatile periods of the experiment. Again, we tested the robustness of these findings with respect to our analysis choices. We found that either applying a more aggressive correction of slow drifts or adopting a trial definition which retained more trials per condition revealed significant interaction effects equivalent to those seen in study 1 under the original pipeline (**Figure 3—figure supplement 1**).

## Biperiden particularly affects mismatch responses during stability

In the ERPs at the sensors which showed significant mismatch stability interaction effects in study 1, it appeared that the interaction effect was mainly driven by the biperiden group (**Figure 3B**). Indeed, when examining the drug groups separately, the interaction effect was significant only in the biperiden group (208 ms, $t$ = 4.96, p = 0.009) at right central channels, but not in the placebo or the amisulpride group. However, there were no clusters for the three-way interaction with drug group which survived multiple comparison correction across the whole time × sensor space. The same held when zooming in on those clusters that showed significant interaction effects, using the functionally defined mask

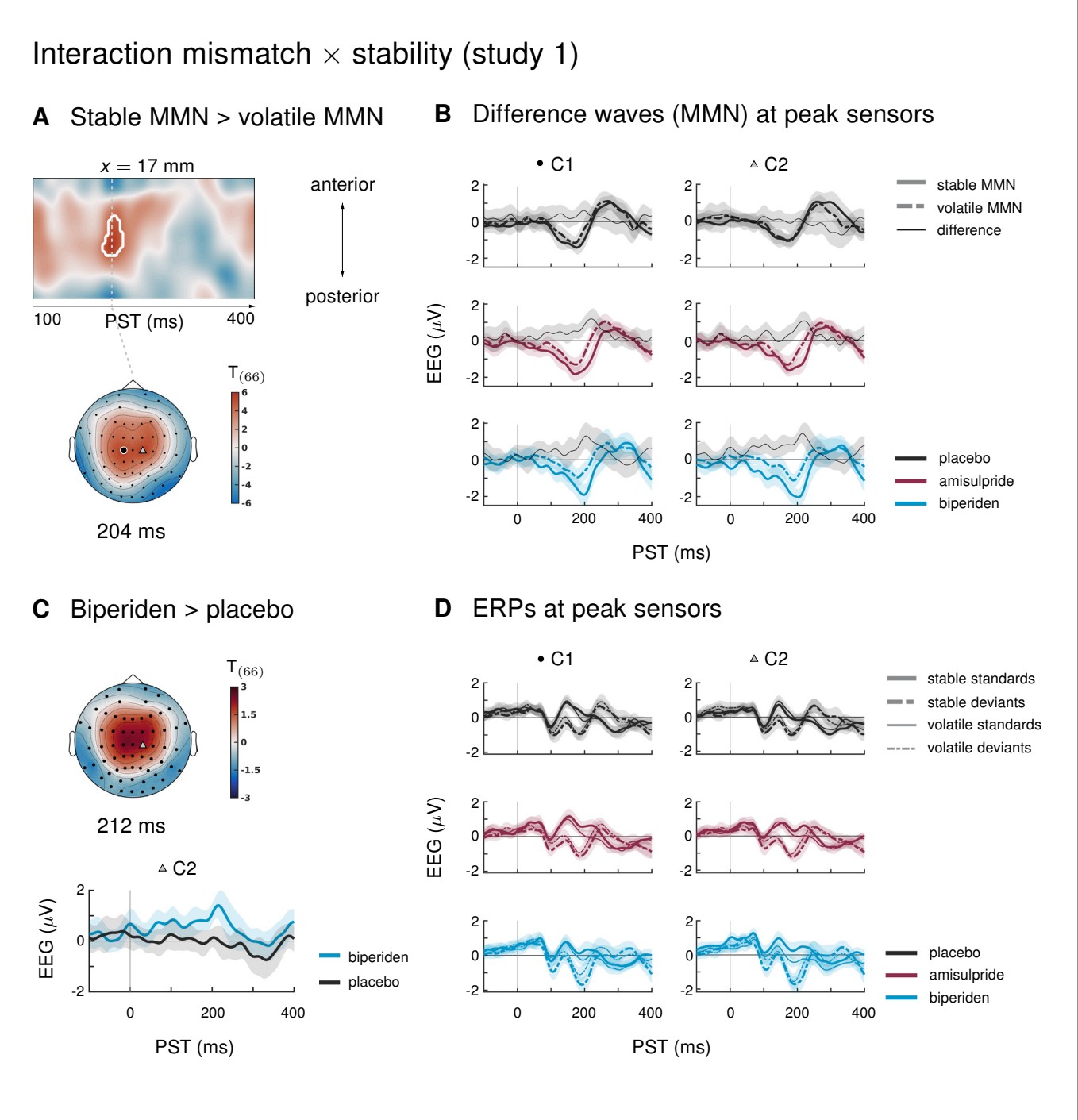

**Figure 3.** Interaction effects between mismatch and stability, and three-way interaction mismatch × stability × drug in study 1. (**A**) Regions of the time × sensor space where ERPs to tones in stable phases were more positive than ERPs to tones in volatile phases. Logic of display as in *Figure 2*. (**B**) ERP difference waves (deviants – standards) at the peak sensors for the two clusters shown in panel A, separately for the three drug groups. (**C**) Pharmacological effect on the interaction: at right central channels, the biperiden group showed a stronger interaction effect between mismatch and stability than the placebo group (significant only within a spatio-temporal mask, see main text). Displayed is the *t*-map of the contrast and the difference waves (volatile – stable MMN) at sensor C2. (**D**) ERPs to standards and deviants at the same sensors as plotted in A and B.

The online version of this article includes the following figure supplement(s) for figure 3:

**Figure supplement 1.** Interaction effect mismatch × stability at a left central sensor across different analysis pipelines.

**Table 5.** Significant clusters of activation for pharmacological effects on stable mismatch in study 1. Columns are organized as in *Table 1*.

| Study 1: Stable Mismatch | cluster | $x$ [mm] | $y$ [mm] | $z$ [ms] | $t_{66}$ | $Z_\equiv$ | $p_{FWE}$ | $k_E$ | $tw_{sig}$[ms] |
|---|---|---|---|---|---|---|---|---|---|
| **A** PLA >BIP | 1 | –26 | 56 | 204 | 4.56 | 4.24 | 0.027 | 8 | 204–212 |
| | 2 | –26 | 61 | 224 | 4.36 | 4.07 | 0.049 | 1 | 224–224 |
| **B** BIP > PLA | 1 | 21 | –19 | 212 | 4.92 | 4.52 | 0.009 | 78 | 200–220 |
| **C** AMI > BIP | 1 | 17 | 72 | 164 | 4.74 | 4.38 | 0.016 | 11 | 160–168 |

of the average interaction effects. However, focusing on only those parts of the time × sensor space where there was a significant positive interaction between mismatch and stability (cluster 1 in *Table 4*), we did find a significant three-way interaction such that the interaction of mismatch and stability was stronger in the biperiden group compared to the placebo group at 212 ms at right central sensors ($t$ = 3.18, p = 0.034 after small volume correction, see *Figure 3C*). Note that, similar to the constrained mask for the overall mismatch effects, this constrained mask was not part of our a priori analysis plan.

In line with our analysis plan, we also examined the interaction of drug with mismatch during stable phases separately from mismatch during volatile phases. Because mismatch effects were stronger during stable periods of the experiment (see above), we suspected that we might also be more sensitive to the effects of the pharmacological manipulation in these periods.

Indeed, while there were no significant effects of drug group on mismatch in volatile phases, drug groups did differ significantly in their mismatch response during stable periods. Again, as for overall mismatch, pre-frontal sensors showed significantly reduced mismatch responses between 160 ms and 168 ms after tone onset in the biperiden group compared to the amisulpride group, peaking at 164ms ($t$ = 4.74, p = 0.016). Additionally, later mismatch responses were significantly larger in the biperiden group compared to placebo at right central and centro-parietal sensors (see *Table 5* and *Figure 2D*), again reflecting a delayed mismatch response under biperiden with a shift in topography from left frontal and pre-frontal toward right central and centro-parietal channels.

When constraining the search volume using the average effect of stable mismatch, the delayed mismatch in the biperiden group was additionally significantly stronger than in the amisulpride group at 204 ms in left pre-frontal sensors ($t$ = 4.17, p = 0.041). Overall, the effects of biperiden on stable mismatch resembled the ones on overall mismatch signals, but with higher effect sizes, while there were no significant pharmacological effects on volatile mismatch.

In study 2, the three drug groups did not differ in how mismatch ERPs were affected by the stability of the current context (three-way interaction with factor drug). Examining mismatch responses in stable and volatile phases separately did not reveal any significant interactions with the factor drug either.

## Discussion

Above, we presented results from two pharmacological EEG studies which were designed to test the sensitivity of a new auditory oddball paradigm to cholinergic and dopaminergic modulations of synaptic plasticity. In study 1, we found that biperiden, a selective muscarinic (cholinergic) M1 receptor antagonist, delays and topographically shifts mismatch responses in this paradigm, while we did not observe this effect when inhibiting dopaminergic D2/3 receptor transmission by administration of amisulpride. Neither elevated cholinergic nor dopaminergic transmission, as induced in study 2 by galantamine and levodopa, respectively, resulted in observable changes to deviance processing in our task.

Our paradigm allowed us to examine processing of auditory deviants in two different contexts: during stable phases of the experiment, one tone reliably served as the 'deviant' (i.e., the unlikely) event, and the other as the 'standard'. During volatile phases, the roles of standard and deviant switched more rapidly, requiring faster updating of the internal model of the acoustic environment. We found that antagonizing muscarinic cholinergic receptors with biperiden affected deviance processing

particularly during stable phases of our task along with a significant interaction between deviance and stability.

In the following, we will first discuss the pharmacological effects on mismatch responses in the light of previous literature, then examine the influence of environmental volatility on mismatch processing and how this interacted with biperiden. Finally, we will discuss the clinical implications of our findings for treatment selection in schizophrenia.

## Delayed and topography-shifted mismatch responses under biperiden

In study 1, mismatch responses in the biperiden group peaked later and were distributed more towards right centro-parietal channels than in the other drug groups (*Figure 2B*). This resulted in significantly smaller mismatch amplitudes at pre-frontal sensors early on, in classical MMN time windows (biperiden vs. amisulpride), and significantly larger MMN at centro-parietal sensors later (biperiden vs. placebo).

One might wonder whether the early difference between the biperiden and the amisulpride group at pre-frontal sensors is difficult to interpret, given the lack of differences of either drug group compared to placebo. However, given our research question – that is, whether auditory mismatch signals are differentially susceptible to muscarinic versus dopaminergic receptor status – showing a significant difference between biperiden and amisulpride is critical.

Clearly, such a differential effect would be even more compelling if biperiden differed significantly from amisulpride and placebo at the same time (and in the same sensor locations). While we do not find this in our main analysis, we do see it for the analysis using the alternative pre-processing pipeline and the trial definition (*Figure 2—figure supplement 3*) that was also specified a priori in our analysis plan. In this alternative analysis, mismatch responses under biperiden did differ significantly from both placebo and amisulpride.

Effects of cholinergic agents on MMN have been demonstrated repeatedly, mostly showing enhanced mismatch amplitudes or shortened MMN latencies in response to stimulation of nicotinic cholinergic receptors (*Baldeweg et al., 2004*; *Harkrider and Hedrick, 2005*; *Dunbar et al., 2007*; *Martin et al., 2009*; *Knott et al., 2012*; *Hamilton et al., 2018*). In contrast, previous investigations of the effects of antagonizing muscarinic cholinergic receptors have yielded less consistent results. Studies using the muscarinic antagonist scopolamine have reported reductions of MMN amplitudes (*Pekkonen et al., 2001*), no effects on MMN (*Pekkonen et al., 2005*), and reduced P300 responses to targets in active oddball tasks (*Meador et al., 1989*; *Curran et al., 1998*; *Brown et al., 2015*). Here, we used a passive auditory oddball task, following the recommendation that MMN assessment is optimal when the participant's attention is directed away from the auditory domain (*Näätänen, 2000*), and tested the effects of biperiden.

Biperiden differs from scopolamine in the specificity of its binding affinity: it has about tenfold higher affinity for M1 as compared to M2–M5 receptors (*Bolden et al., 1992*). Two studies have tested the effects of biperiden on deviance detection in passive auditory oddball tasks (*Klinkenberg et al., 2013*; *Caldenhove et al., 2017*). Neither study found effects on MMN, but hints of a potential effect on P3a amplitudes. When adopting their pre-processing strategy, our analysis revealed a similarly late effect on mismatch responses that survived whole volume correction (see *Figure 2—figure supplement 3*). Importantly, both previous studies used only half the dose (2 mg) of biperiden as administered here, which might explain the difference in findings compared to our study.

Most previous pharmacological studies on auditory mismatch restricted their examination of drug effects to specific sensors and time points, mostly based on average mismatch difference waves centered on the MMN response. Here, we provide a characterization of the drug effect on mismatch responses across the full time × sensor space, while correcting for multiple comparisons across time and sensors. This analysis revealed both a delay in peak mismatch amplitude, and a shift in topography in the biperiden group (*Figure 2*). Importantly, this shift affects traditional MMN sensors (Fz, FCz, Cz), which have mostly been examined in previous studies, less than those at the border of the MMN scalp distribution (Fp1, Fp2, C2, C4), which is where we found significant effects of biperiden. Another strength of our study design – with total N = 162 across studies – was the use of individual drug plasma level estimates in the group level GLM, based on the analysis of blood samples, which allowed us to account for interindividual differences in pharmacokinetics. We further controlled for potential confounds by means of our inclusion criteria, for example, excluding smokers to avoid effects of baseline nicotine levels. The focus on male participants was intended to avoid confounds of

fluctuating estrogen levels, which have been found to significantly impact on dopaminergic and cholinergic systems (*Gasbarri et al., 2012*; *Colzato and Hommel, 2014*; *Barth et al., 2015*). However, this also constitutes a significant limitation of our study, as it means that our results may not equally apply to both sexes and will therefore need to be replicated in a more representative sample in future work.

Surprisingly, in study 2, we did not find an effect of galantamine on mismatch responses. This is in contrast to a previous report showing an augmentation of MMN under the same dose of galantamine as administered here (*Moran et al., 2013*). The study by Moran and colleagues employed a 'roving' oddball paradigm (*Garrido et al., 2008*): a tone sequence comprised mini-blocks of 6–10 tone repetitions, where consecutive mini-blocks differed in frequency, and the first tone of a block represented the deviant. Importantly, in their paradigm, every deviant indicated the onset of a new context, and contexts (mini-blocks) lasted for at least six tones. In contrast, in our paradigm, mini-blocks of repeated tones tended to be much shorter (less than 20% of mini-blocks with 6 repetitions or more, more than 60% consisting of 2 repetitions or less), making our paradigm considerably more volatile overall. Speculatively, this comparably high tonic volatility could mean that precision-weights on PEs were already high; this might have prevented any increase in sensory precision afforded by galantamine to be expressed in the mismatch ERPs in our study due to a ceiling effect.

Notably, our analysis differed from previous investigations of the effects of biperiden and galantamine on the MMN (e.g. *Klinkenberg et al., 2013*; *Moran et al., 2013*; *Caldenhove et al., 2017*) in terms of pre-processing choices such as the choice of reference, the amount of high-pass filtering and the application of baseline correction. To examine the robustness of our main results to these analysis choices and to rule out that differences in results between studies were simply due to different pre-processing strategies, we re-analyzed our data using equivalent settings to these previous reports. The results of this analysis were largely supportive of our claims: we also found significant effects of biperiden (versus placebo and amisulpride) on mismatch signals that were compatible with an increase in MMN latency, even when adopting the same pipeline as used in previous studies. Similarly, there was still no effect of galantamine on mismatch responses in a whole time × sensor space analysis based on this adapted pipeline.

## Biperiden and the influence of environmental volatility on mismatch processing

In classical oddball paradigms, the occasional deviant represents a rule violation, but its impact on subsequent rule representation is limited, as the tone sequence reverts back to the standard tone immediately. In contrast, the roving oddball paradigm examines model *updating* in a changing environment, as every deviant signals the onset of a new rule. In our new volatility oddball paradigm, the relevance of the detected rule violation to the representation of the rule additionally varies across different periods of the experiment: during more stable phases, oddballs represent noise and deviants should not lead to a major update of the current belief about the underlying rule. In contrast, during volatile phases, the probabilities of the two tones sometimes reverse and deviants thus occasionally signal the onset of a new rule. Theoretical treatments suggest that this volatility can impact on the size of belief updates in two opposing ways (*Mathys et al., 2011*). On the one hand, increased belief uncertainty due to environmental volatility should increase learning rates (i.e., belief updates) – in other words, deviants are more meaningful in volatile phases due to the occasional rule switch. On the other hand, stable phases allow for a more precise prediction of the input than volatile phases, as beliefs about the more likely tone occurrence are allowed to accumulate for longer. This suggests an increased impact of deviants during stability. It is a priori not clear which of these two opposing effects would dominate in a given setting. In our case, we examined this question by contrasting mismatch effects between stable and volatile periods of our task.

In study 1, we found a significant difference between stable and volatile mismatch responses, such that mismatch was stronger in stable than in volatile periods (*Figure 3*). This mirrored previous reports of volatility effects on mismatch signals (*Todd et al., 2014*; *Dzafic et al., 2020*). However, in our study, this was mainly due to the altered mismatch response in the biperiden group, which was particularly affected during stable mismatch. Neither the placebo group nor the amisulpride group showed interaction effects on their own, and, when directly contrasting the groups and focusing on the cluster of central channels that showed the average effect across drug groups, the effect was significantly stronger in the biperiden group compared to placebo. No significant differences between stable

and volatile mismatch responses were found in study 2. Note however, that under a different pre-processing pipeline (with more aggressive correction of slow drifts) and/or trial definition (optimized for retaining more trials per condition), we did see these interaction effects more robustly across all drug groups in both studies (*Figure 3—figure supplement 1*).

It should also be noted that previous reports have presented effects of block-wise volatility changes on mismatch processing in single-channel analyses (focusing only on Fz, *Todd et al., 2014*; *Dzafic et al., 2020*), and that the whole-volume corrected effect presented in *Dzafic et al., 2020* did not replicate in a validation data set, suggesting that the effects of block-wise volatility on mismatch might be relatively subtle compared to the size of the mismatch effect itself (and thus require more aggressive pre-processing or higher trial numbers to be robustly detected). This subtlety of block-wise stability variation might also be explained by the two opposing effects of volatility on precision-weights described above, which might cancel each other to some degree.

We have previously capitalized on the history-dependence of EEG amplitudes in the MMN paradigm where trial-wise amplitude changes carry information about the temporal dynamics of the belief updating process (*Lieder et al., 2013a*; *Stefanics et al., 2018*; *Weber et al., 2020*). This is particularly relevant for considering the impact of environmental volatility on learning rates (*Behrens et al., 2007*; *Mathys et al., 2011*), for example using ideal Bayesian observer analyses (*Stefanics et al., 2018*; *Weber et al., 2020*) due to the passive nature of the paradigm. The results of an equivalent analysis for the current dataset (presented in Appendix 1) were highly consistent with our previous reports applying the same model to other MMN paradigms (*Stefanics et al., 2018*; *Weber et al., 2020*), in that we found multiple, hierarchically related prediction errors underlying EEG mismatch signals: trial-wise EEG amplitudes of the classical MMN component correlated with lower-level prediction errors about tone probabilities, while later P3-like components scaled with a higher-level prediction error about environmental volatility.

Importantly, in line with the results from the conventional averaging approach, we found that only biperiden affected the EEG signatures of model-derived precision-weighted prediction errors – neither dopaminergic manipulations nor galantamine showed significant differences to the placebo group. Moreover, the biperiden effect concerned not only signatures of low-level prediction errors, reflecting mismatches between expected and actual tone identities, but also signatures of higher level prediction errors serving to refine beliefs about environmental volatility. However, for ease of accessibility and to enable direct comparison with previous work on dopaminergic and cholinergic effects on the MMN, we have focused our conclusions on the results of the conventional ERP analysis, and only present the model-based analysis in the appendix.

## Future directions

In this study, we employed a conventional ERP analysis, but considered all sensors and time points under multiple comparison correction, to detect effects of experimental conditions that manifest as differences in evoked response amplitudes within our time-window of interest. In our main analysis, the effect of biperiden on mismatch signals as compared to placebo appeared relatively subtle but was retrieved with higher effect sizes and survived multiple comparison correction across the whole time × sensor space when *(a)* focusing on the stable phases of our experiment (in line with our analysis plan, *Figure 2D*), despite the reduction in trial numbers this entails, *(b)* using an alternative pre-processing pipeline with more aggressive correction of slow drifts and retaining more trials per condition (*Figure 2—figure supplement 3*), or *(c)* adopting a model-based approach which takes into account all trials of the experiment (Appendix 1). Similarly, the effect of environmental volatility showed up more robustly across studies when either using the alternative pre-processing, or retaining more trials per condition, or both (*Figure 3—figure supplement 1*). This suggests that the pre-processing and statistical strategy of the main analysis did not have optimal sensitivity for detecting these effects. Thus, for future studies using our paradigm, either focusing the analysis on an ROI based on the average effects presented here, or adopting the model-based analysis approach, would be promising strategies – in particular, when extracting effects in individual participants or patients.

Furthermore, our pattern of results – an apparent biperiden-induced shift in mismatch responses from an early to a later peak, and from frontal to central channels – suggests that methods which go beyond the amplitude-based approach used here and exploit the rich temporal information in the EEG signal could help us to further understand the impact of cholinergic neurotransmission on

perceptual inference in our task. Examples for this are principal component analysis (PCA) based analyses (*Hunt et al., 2015*), which take into account the topography as well as the time course of the ERP, or dynamic causal modeling (DCM), which interprets scalp-level effects in terms of extrinsic (between-area) connectivity changes and local effects (such as synaptic gain modulation within an area) in an underlying network of sources (*David et al., 2006*; *Kiebel et al., 2006*; *Garrido et al., 2007*). Future analyses of the current data set might employ this technique to infer on low-level (synaptic) mechanisms underlying the observed pharmacological effects, for example, biperiden-induced changes in post-synaptic gain of supragranular pyramidal cells in auditory cortex (*Moran et al., 2013*; *Schöbi et al., 2021*).

## Clinical implications

Together, the current analyses demonstrate that mismatch responses in our paradigm were sensitive to muscarinic receptor status. In contrast, and in line with previous reports (*Kähkönen et al., 2002*; *Leung et al., 2007*; *Leung et al., 2010*; *Korostenskaja et al., 2008*), dopaminergic challenges in both of our studies did not affect mismatch responses. Notably, in our control analyses, biperiden differed significantly from both placebo and amisulpride in the same sensors and time points. Such a differential sensitivity to cholinergic versus dopaminergic neuromodulation may prove valuable for understanding and predicting differential treatment responses in individuals diagnosed with schizophrenia. Importantly, while the reduction of MMN amplitudes in patients compared to healthy controls is robust and of large effect size (*Erickson et al., 2016*), there is still considerable inter-individual variation in MMN amplitudes among patients (*Light and Swerdlow, 2015*), supporting the idea that different subgroups of patients might differ in their MMN expression. Based on our results, we speculate that reduced MMN in patients might be relatively more indicative of cholinergic versus dopaminergic dysregulation of synaptic plasticity.

Notably, there is compelling evidence for a subgroup of patients with markedly decreased M1 receptor availability in the prefrontal cortex (*Scarr et al., 2009*, see also *Gibbons et al., 2013* and *Scarr et al., 2018*). This is consistent with the possibility that a key pathophysiological dimension of the heterogeneity of schizophrenia derives from differential impairment of cholinergic versus dopaminergic modulation of NMDAR function (*Stephan et al., 2006*; *Stephan et al., 2009*).

Distinguishing these potential subtypes of schizophrenia could be highly relevant for treatment selection, as some of the most effective neuroleptic drugs (e.g., clozapine, olanzapine) differ from other atypical antipsychotics (e.g., amisulpride) in their binding affinity to muscarinic cholinergic receptors. The exact mechanisms by which muscarinic receptors are involved in the therapeutic effects of clozapine and olanzapine are still under debate and include, for example, elevation of extracellular levels of acetylcholine in cortex (*Ichikawa et al., 2002*; *Shirazi-Southall et al., 2002*; *Weiner et al., 2004*), possibly via blocking presynaptic muscarinic autoreceptors (see *Johnson et al., 2005*; *Tzavara et al., 2006* for conflicting data), and normalization of M1 receptor availability in cortex (*Malkoff et al., 2008*).

Irrespective of the exact mechanism by which clozapine and olanzapine exert their antipsychotic effects, their much higher affinity to muscarinic cholinergic receptors compared to dopaminergic receptors sets them apart from other antipsychotics. If a functional readout of the relative contribution of cholinergic versus dopaminergic deficits could be obtained in individual patients, this might be predictive of whether this patient would profit from clozapine, olanzapine, or, in the future, potential new treatments targeting the muscarinic system specifically. Indeed, muscarinic receptors have become an important target of drug development for schizophrenia (*Yohn and Conn, 2018*).

To establish the utility of our paradigm in the clinical context, two things would be important. First, the test-retest reliability of the proposed biomarker. Generally, previous studies have shown promising results for the MMN, with high intra-class correlation coefficients (up to > 0.90) and other stability measures in both healthy and clinical populations (*Light et al., 2012*; *Roach et al., 2020*; *Wang et al., 2021*), consistent with the idea that ERPs tend to be idiosyncratic in their expression, but reliable within individuals (*Gaspar et al., 2011*). However, a dedicated test-retest reliability study for the specific MMN variant in this article has not yet been conducted.

Second, prospective patient studies are needed, which test whether this readout of cholinergic neurotransmission is predictive of treatment success in individual patients. In particular, such a prediction may become possible by adopting the 'generative embedding' strategy frequently used in

translational neuromodeling and computational psychiatry (*Stephan et al., 2017*): this involves estimating synaptic variables of (generative) neuronal circuit models of MMN and using these estimates as features for subsequent machine learning. While the potential of this computational strategy, in the specific context of muscarinic manipulations of the MMN, was demonstrated by a recent rodent study (*Schöbi et al., 2021*), an important question for future work is whether it can be successfully translated to a clinical setting.

## Materials and methods
### Analysis plan, data and code availability
Prior to the unblinding of the researcher conducting the analysis, a version-controlled and time-stamped analysis plan was created. This plan detailed the analysis pipeline ex ante (see next sections). The analysis plan is provided online at https://gitlab.ethz.ch/tnu/analysis-plans/weber-muscarinic-mmn-erp. The data used for this manuscript are available at https://research-collection.ethz.ch/handle/20.500.11850/477685, adhering to the FAIR (Findable, Accessible, Interoperable, and Re-usable) data principles. Furthermore, the analysis code that reproduces the results presented here is publicly available on the GIT repository of ETH Zurich at https://gitlab.ethz.ch/tnu/code/weber-muscarinic-mmn-erp-2021. The code used for running the experimental paradigm will also be made publicly available, as part of a future release of the open source software package TAPAS (https://www.translationalneuromodeling.org/tapas).

### Study 1
#### Participants
In total, 81 volunteers (mean age 22.7 years (SD = 3.6, range = 18–38)) participated in study 1. The data reported here were collected as part of a larger project which included other paradigms and data modalities. Sample size per study was chosen to obtain a statistical power of 80% for detecting significant differences between drug conditions (assuming a significance threshold of $\alpha$ = 0.05, an expected drop-out rate of 25%, and an effect size of d = 0.8 or larger). In this initial study with its focus on the feasibility of an EEG-based readout of differential sensitivity to cholinergic (muscarinic) vs. dopaminergic function, we aimed for controlling potential confounds as tightly as possible. In addition to measuring individual drug plasma levels and transmitter-relevant single-nucleotide polymorphisms (see below), we therefore only recruited male participants in order to avoid the significant influence of fluctuating estrogen levels on dopaminergic and cholinergic systems (*Gasbarri et al., 2012*; *Colzato and Hommel, 2014*; *Barth et al., 2015*). However, this has the obvious disadvantage that our study is not representative for the entire population. This is a significant limitation which we revisit in the Discussion. All participants were right-handed, Caucasian, and non-smokers with normal or corrected-to-normal vision. Further exclusion criteria included serious chronic or current physical or mental illness, drug consumption, and hearing aids.

To exclude any cardiac abnormalities that could render a pharmacological intervention risky, participants underwent a clinical examination including electrocardiogram (ECG) before data acquisition. Participants were randomly assigned to one of three drug groups: placebo, amisulpride, or biperiden (between-subject design, N = 27 per group), with both the participant and the experimenters blind to the drug label. All participants gave written informed consent prior to data acquisition and were financially reimbursed for their participation. The study was approved by the cantonal Ethics Committee of Zurich (KEK-ZH-Nr. 2011-0101/3).

Data from a total of 10 participants could not be used in the group analysis presented here for the following reasons: change of the stimulus sequence after the first few participants (N = 6), technical issues during measurement (N = 2), and failure to sufficiently correct for eye blink artefacts during preprocessing of EEG data (N = 2, see below). Therefore, the results reported here are based on a final sample of N = 71 participants, with N = 25 in the placebo group (mean age 23.2 years [SD = 4.8, range = 18–38]), N = 24 in the amisulpride group (mean age 22.4 years [SD = 3.4, range = 18–33]), and N = 22 in the biperiden group (mean age 22.5 years [SD = 3.1, range = 18–29]). Criteria for excluding data sets from the group analysis were defined and documented in a time-stamped analysis plan prior to un-blinding of the analyzing researcher (see below, section 'Analysis Plan, Data and Code Availability').

## Pharmacological substances and administration

At the clinical examination, participants were instructed to abstain from the consumption of alcohol and grapefruit juice for 24 hr before the EEG measurement, not to take any medications within 3 days before the experiment and not to consume other drugs. They were further instructed not to eat for 3 hr before the EEG measurement, and to abstain from driving a car for 48 hr after the experiment.

Approximately 80 min before the start of the EEG measurement, capsules of each compound (amisulpride/biperiden/placebo) were administered as a single oral dose. All capsules had the same visual appearance and drug administration was conducted in a double-blind fashion. The drugs were prepared by the local pharmacy Bellevue Apotheke, Zurich.

Amisulpride was administered using Solian 400 mg mixed with 570 mg of lactose. At this dose, amisulpride blocks postsynaptic $D_2$ and $D_3$ receptors, thus inhibiting DA transmission (*Chhabra and Bhatia, 2007*). Biperiden capsules contained two units of 2 mg Akineton (i.e., 4 mg in total) mixed with 880 mg of lactose. Biperiden is the most selective M1 antagonist available for human subjects (*Katayama et al., 1990*; *Bolden et al., 1992*) and has only minor peripheral anticholinergic effects in comparison with other anticholinergic substances. Placebo capsules contained 960 mg of lactose.

## Blood samples

Four blood samples were collected per participant in order to (1) estimate the actual drug plasma levels at the time participants performed the experimental task (using two samples), and (2) to assess genetic variation at functional single nucleotide polymorphisms (SNPs) of two genes relevant to the pharmacological intervention (using two samples). However, the assessment of genetic effects in our study is constrained by the very limited sample size. In particular, for some genotypes of interest, there were only 2 or 3 individuals within certain drug groups showing these genotypes. We therefore refrain from interpreting or discussing these genetic effects any further and report them in Appendix 3 for completeness and potential guidance for future follow-up studies with larger sample sizes.

## Drug plasma concentration

For both pharmacological agents, the expected maximal plasma concentration was around 1 hr after intake (amisulpride: first peak of plasma concentration after 1 hr, second peak at 3–4 hr, absolute bioavailability of 48%, elimination half-life ~12 hr (https://compendium.ch/mpro/mnr/8962/html/de); biperiden: for single dose usage, peak of plasma concentration around 1 hr after administration, absolute bioavailability ~33%; elimination half-life 11–21.3 hr (https://compendium.ch/mpro/mnr/1853/html/de)).

The first blood sample was collected on average 75.67 min (SD = 3.22) after drug intake. A second blood sample was taken on average 188.99 min (SD = 9.91) after drug administration. Blood samples were collected in tubes containing heparin as anticoagulant, centrifuged at 10 °C for 10 min at 3000xg and finally stored at –86 °C until analysis.

Blood analysis was performed by the Institute of Clinical Chemistry at the University Hospital Zurich with a detection threshold of 1 nmol/L. Samples were measured using liquid chromatography coupled to tandem mass spectrometry (LC-MS/MS). Methods were fully validated and accredited according to ISO 17025. The lower limits of quantification were for amisulpride 2 nmol/L, for biperiden 1 nmol/L, for galantamine 1 μg/L, and for levodopa 10 μg/L.

Estimated drug plasma levels at the time of the experimental task were read off a linear approximation of drug concentration decay between the two collection time points for each individual and entered the group level general linear model (GLM) as a covariate (see below).

## MMN paradigm

Participants passively listened to a sequence of tones, presented binaurally through headphones, while engaging in a visual distraction task (described below). The auditory stimuli consisted of two pure sinusoidal tones; a high (528 Hz) and a low (440 Hz) tone. A total of 1800 tones were presented, with a duration of 70 ms each and an inter-stimulus interval of 500 ms, see *Figure 1* for a visualization of the paradigm and relative timing of events. Auditory and visual stimuli were presented using Psych-Toolbox (PTB3, psychotoolbox.org).

The probability of hearing the high tone was either 0.15 (in which case it was the deviant) or 0.85 (in which case it functioned as a standard), except for four short phases, 50 trials each, in which the

probability of hearing either tone was equal. In order to ensure that both tones appeared equally often in both roles, the second half of the stimulus stream was a repetition of the first half, with only the tones switched. This avoids potential confounding effects by ensuring that both stimulus categories have, on average, the same physical properties across the duration of the experiment.

### Visual distraction task

Participants performed a distracting visual task and were instructed to ignore the sounds. The task consisted of detecting changes to a centrally presented small white square. Whenever the square opened to either the left or the right side, participants were instructed to press a button on a response box with their index finger (left opening) or middle finger (right opening). The 36 'square openings' (half of them to the left) occurred at irregular intervals and did not coincide with tone presentations but always followed a tone with a delay varying randomly between 50 and 250 ms after tone offset (see *Figure 1B*).

### EEG data acquisition and preprocessing

EEG data were collected at a sampling rate of 500 Hz using an EASYCAP system 64 scalp electrodes including one electrooculography (EOG) channel (10–20 layout; EASYCAP GmbH, https://www.easycap.de/wordpress/). Data were recorded with nose-reference. Before starting the experimental task, impedances were ensured to be well below 20 kOhm for all channels. For a subset of participants, ECG and pulse oximetry data were additionally acquired via a bipolar amplifier (BrainAmp ExG; Brain Products GmbH, https://www.brainproducts.com/index.php); however, these data were not analyzed in the present study. For one participant, erroneous cabling during data acquisition resulted in a different order of EEG channels. This could be corrected for during the pre-processing of the data.

Pre-processing and data analysis of EEG data were performed using SPM12 (v6906, http://www.fil.ion.ucl.ac.uk/spm/) and Matlab (R2018b). Continuous EEG recordings were re-referenced to the average, high-pass filtered using a Butterworth filter with cutoff frequency 0.1 Hz, down-sampled to 250 Hz, and low-pass filtered using Butterworth filter with cutoff frequency of 30 Hz.

The data were epoched into 500 ms segments around tone onsets, using a pre-stimulus time window of 100 ms. We did not baseline-correct the epochs. Whether the benefits of traditional baseline correction outweigh its downsides is still a matter of debate (*Alday, 2019*). Here, we wanted to avoid mixing anticipation or prediction signals with event-related responses, which we interpret as learning or model update signals. However, to explore the robustness of our findings under different choices of analysis strategy, we also present results under an alternative pre-processing pipeline which included strong high-pass filtering and baseline correction (section Control analyses).

A vertical EOG channel was computed as the difference between channel Fp1 and the EOG channel which was placed beneath the left eye. We accounted for eye-movement related artefacts by applying the signal space projection (SSP) eye blink correction method (*Nolte and Hämäläinen, 2001*) as implemented in SPM12: This approach uses an estimate of the spatial topography due to eye activity to define ocular source components and removes eye activity by regressing these components out of the EEG data.

In particular, eye blink events were identified with a thresholding approach applied to the data from the vertical EOG channel. Detected eye blink events were used to epoch the continuous EEG into 1000 ms segments around these events, excluding any epochs containing large transients. Ocular components were determined using singular value decomposition (SVD) of topographies from all the eye blink trials and all the time points. The leading SVD component was used to define the noise subspace that was subsequently projected out of the data (*Nolte and Hämäläinen, 2001*). This projection was applied to the data epoched around the auditory stimulus presentation. For all participants, we verified that the leading SVD component had the typical spatial topography of an eye blink artifact and resulted in satisfactory eye blink correction performance (inspected visually by plotting the average eye blink in a subset of channels before correction and after correction). To achieve this, in a subset of participants, the default eye blink detection threshold of 5 SD was changed to a value that resulted in improved correction performance. Participants for which such a component could not be identified were excluded from further analysis (N = 2: one in the amisulpride group, one in the biperiden group).

Finally, epochs in which the (absolute) signal recorded at any of the channels exceeded 75 μV were removed from subsequent analysis. For all channels in all participants, the number of excluded epochs was below 20% of the total number of epochs. The number of remaining good trials was 1,775 on average across participants (SD = 28) and almost identical across drug conditions (placebo group: 1775, SD = 29; amisulpride group: 1775, SD = 27; biperiden group: 1776, SD = 30).

The remaining good trials were converted, for each participant, into scalp images for all 63 EEG channels and all time points between 100 ms and 400 ms after tone onset, using a voxel size of 4.2 mm × 5.4 mm × 4.0 ms. The images were spatially smoothed with a Gaussian kernel (FWHM: 16 mm × 16 mm) in accordance with the assumptions of Random Field Theory (*Worsley et al., 1996*; *Kiebel and Friston, 2004*) and to accommodate for between-subject spatial variability in channel space. To avoid confusion, we only use the term 'MMN' when we talk about effects in the classical time window (100–200 ms) and sensor locations (frontocentral sensors) for the MMN, and use 'mismatch responses' for all other effects.

## First-level general linear model

We defined categorical trial types based on our tone sequence: deviant trials were defined as the first tone with a different frequency; following previous studies (*Garrido et al., 2008*), we only considered tones as deviants which followed at least 5 repetitions of the other tone (N = 119). Equivalently, standard trials were defined as the 6th repetition of the same tone (N = 106) in order to keep trial numbers comparable across conditions. Based on the probability structure of the input sequence, we further divided these into deviants in stable phases, deviants in volatile phases, standards in stable phases, and standards in volatile phases. Stable phases were defined as phases in which the probability of hearing the high tone did not change for at least 100 trials; volatile phases were all other phases of the experiment. An alternative trial definition was applied in the analyses examining robustness (section Control analyses).

Per participant, we modeled the trial-wise 3D ERP images with a GLM which implements a factorial design with two factors: 'Mismatch' (levels: 1. Standards, 2. Deviants) and 'Stability' (levels: 1. Stable, 2. Volatile). With regard to non-sphericity correction at this single-subject level, we assumed that the error might have different variance (i.e. non-identity) but is not correlated (independence) across conditions, in line with the recommendations in the SPM manual (http://www.fil.ion.ucl.ac.uk/spm/). This GLM only served to provide the contrast images to be used in the group level GLM. We computed contrast images (using *t*-tests) for the following contrasts of interest:

- mismatch effect: standards vs. deviants
- stability effect: stable vs. volatile
- interaction effect: stable mismatch vs. volatile mismatch
- stable mismatch: stable standards vs. stable deviants
- volatile mismatch: volatile standards vs. volatile deviants

For visualization purposes, grand average waveforms were computed for each condition.

Additionally, and in line with our analysis plan, we performed a model-based analysis in which the conventional trial definition ('standard' versus 'deviant' trials) was replaced with a trial-wise estimate of the amount of prediction error that each tone in our sequence would elicit, according to an ideal observer model. This analysis has the advantage of taking into account the trial-by-trial dynamics; on the other hand, it requires making assumptions about the nature of the learning process. The details of this analysis and the obtained results are presented in Appendix 1.

## Group level general linear models

Random effects group analysis across all participants was performed using a standard summary statistics approach (*Penny and Holmes, 2007*). We used a separate group-level GLM for each effect of interest from the first level GLM, which implements a factorial design with the between-subject factor 'drug' (levels: 1. Placebo, 2. Amisulpride, 3. Biperiden). With regard to non-sphericity correction, the group-level analysis assumed independence (measurements are unrelated to each other), given the between-subject design, and non-identity (variances may differ across measurements).

We introduced a covariate for the estimated drug plasma concentration levels of both pharmacological agents, where we allowed for an interaction with the drug factor and mean-centered the covariate within drug groups.

In sum, our design effectively comprised two within-subject factors – mismatch (standards vs. deviants) and stability (stable vs. volatile), which we specified in our first-level GLM – and one between-subject factor, drug group (placebo vs. amisulpride vs. biperiden). At the group level, we were particularly interested in the interaction between the factors mismatch and drug, and the three-way interaction between mismatch, stability, and drug.

## Pharmacological effects

For each effect of interest from the first level, we used eight separate t-tests to examine: average positive and negative EEG deflections for the effect across drug groups, and drug differences in the expression of the effect: amisulpride compared to placebo in both directions, biperiden compared to placebo in both directions, and differences between amisulpride and biperiden in both directions.

In addition to an initial analysis across the whole time-sensor space, we investigated drug effects within a smaller, functionally constrained search volume, which comprised those regions of the time × sensor space where we found significant average effects (across drugs). Specifically, a mask was functionally defined for each effect of interest and created by combining the images of significant activations for the positive and the negative average effect (logical OR) of that contrast. Importantly, the differential contrasts used to test for drug effects were orthogonal to the average contrasts used to construct these masks.

For all analyses, we report all results that survived family-wise error (FWE) correction, based on Gaussian random field theory, across the entire volume (time × sensor space), or within the functional masks (small volume correction, SVC), at the peak level (p < 0.05).

# Study 2

Study 2 employed exactly the same study design as study 1 except for the pharmacological agents used. The participants did not overlap across studies. In the following, we only report the parts of the experiment that differed to study 1 and refer the reader to study 1 for all other aspects of the experiment and analysis. In particular, we followed exactly the same analysis steps as outlined in the analysis plan for study 1 (see section Analysis plan, data and code availability).

## Participants

In total, 81 male volunteers (mean age 23.5 years [SD = 3.5, range = 18–35]) participated in study 2. Inclusion and exclusion criteria were identical to study 1.

Participants were randomly assigned to one of three drug groups: placebo, levodopa, or galantamine (between-subject design, N = 27 per drug group) with both participants and experimenters blind to the drug label. All participants gave written informed consent prior to data acquisition and were financially reimbursed for their participation. The study was approved by the cantonal Ethics Committee of Zurich (KEK-ZH-Nr. 2011-0101/3).

Data from three participants could not be used in the group analysis presented here due to a diagnosis of diabetes (N = 1; prior to unblinding, we decided not to analyze this dataset because of potential interactions of insulin with DA; *Figlewicz et al., 2003*; *Fiory et al., 2019*), technical issues during measurement (N = 1), and an adverse event prior to data acquisition (N = 1; nausea). Therefore, the results reported here are based on a sample of N = 78 participants, with N = 26 in the placebo group (mean age 24.3 years [SD = 3.9, range = 19–35]), N = 26 in the levodopa group (mean age 23.6 years [SD = 3.8, range = 19–33]), and N = 26 in the galantamine group (mean age 22.7 years [SD = 3.0, range = 18–33]).

## Pharmacological substances, administration, and blood samples

Approximately 80 min before the start of the EEG measurement, capsules of each compound (levodopa/galantamine/placebo) were administered as a single oral dose. All capsules had the same visual appearance and drug administration was double-blind.

For levodopa, we followed closely the procedure reported by *Rihet et al., 2002* by using a single oral dose administration of Madopar DR (Roche Pharma (Switzerland) AG, 4,153 Reinach; Licence number: 53,493 (Swissmedic)), mixed with 670 mg lactose. Madopar DR is a dual-release formulation containing 200 mg levodopa and 50 mg benserazide. Levodopa is the immediate metabolic precursor of DA and is decarboxylated to DA both in the central (CNS) and the peripheral nervous system.

Concurrent administration of benserazide, a dopa decarboxylase inhibitor, which does not cross the blood-brain barrier, reduces the extracerebral side effects of levodopa and enhances the amount of levodopa reaching the CNS (*Crevoisier et al., 1987*). Galantamine was administered as a single oral dose of Reminyl (Janssen-Cilag (Switzerland) AG, Baar, ZG; Licence number: 56,754 (Swissmedic)) containing 8 mg of galantamine, mixed with 920 mg lactose. As a selective, competitive and reversible inhibitor of acetylcholinesterase (AChE), an enzyme which degrades ACh, galantamine increases the availability of ACh. Additionally, it may act as a positive allosteric modulator of nicotinic receptors (*Schrattenholz et al., 1996*; *Samochocki et al., 2003*) although this property is being debated (*Kowal et al., 2018*). Placebo capsules only contained lactose. Drugs were prepared by the local pharmacy Bellevue Apotheke, Zurich.

For both pharmacological agents, the expected maximal plasma concentration was around 1 hr after intake (levodopa: peak of plasma concentration after 1 hr, absolute bioavailability of ~78% when using the dual-release formulation, elimination half-life ~1.5 hr (https://compendium.ch/product/56931-madopar-dr-tabl-250-mg/mpro); galantamine: peak of plasma concentration around 1–2 hr after administration, absolute bioavailability ~88.5%; elimination half-life 7–8 hr (https://compendium.ch/product/1018816-reminyl-prolonged-release-kaps-8-mg/MPro)).

The first blood sample was collected on average 77.71 min (SD: 14.38) after drug intake. A second blood sample was taken 192.79 min (SD: 18.45) after drug administration. Blood samples were collected and processed as described in study 1. As in study 1, an additional blood sample was collected for assessing genetic variation at selected functional single nucleotide polymorphisms (SNPs). As for study 1, we report the results of the genetic analyses in Appendix 3, bearing in mind the sample size limitations mentioned above.

## Paradigm and distraction task

We used the same paradigm and distraction task as in study 1. However, following observations during study 1 that participants found the task rather tiring due to long sequences without visual events, we increased the number of square openings in the visual distraction task from 36 to 90 to make the task more engaging. One participant had a hit rate below 75% (see section Results). Again, we report the group level results including data from this participant in the main text, but also report the results based on the analysis without this dataset (*Supplementary file 1*).

## EEG recording and statistical analysis

EEG recording setup, preprocessing pipeline and statistical analysis were identical to study 1. For all channels in all participants, the number of excluded epochs was below 20% of the total number of epochs, therefore, we did not mark any channels as bad. The number of remaining good trials was 1753 on average (SD = 71), with no significant differences (one-way ANOVA $F$ = 1.18, p = 0.31) across groups (placebo: 1770, SD = 51; levodopa: 1740, SD = 86; galantamine: 1750, SD = 71). The specification of first level and group level GLMs was identical to study 1.

## Control analyses

In separate analyses, we examined the robustness of our main findings to changes in analysis strategy. We were particularly interested in *(1)* the main effect of mismatch and its modulation by drug group, and *(2)* the interaction of mismatch and stability, after correcting for slow drifts and increasing trial numbers.

First, to make our results more directly comparable with previous reports on the effects of biperiden and galantamine on the MMN (*Klinkenberg et al., 2013*; *Moran et al., 2013*; *Caldenhove et al., 2017*), and address concerns of being not sensitive enough to potentially subtle drug effects, we *copied the pre-processing settings from these reports*, resulting in the following changes to our previous pipeline:

1. A re-referencing to a linked mastoid reference
2. A strong high-pass filter of 1 Hz
3. A baseline correction using a 100 ms pre-stimulus time window

All other pre-processing steps remained as reported in the main analysis.

Second, we used a different standard and deviant definition (that we had already specified as part of our analysis plan): instead of only considering tones as standards and deviants that followed at least 5 repetitions of the same tone (initial, 'INIT' MMN definition), we instead considered every tone after at least 2 repetitions. Because this results in many more standard tones than deviant tones, we only used a subset of these standard tones, choosing them such that for a given number of preceding tone repetitions, there were as many standards as there were deviants (new, '2REP' MMN definition; note that in the publicly available analysis plan, these definitions were labeled 'roving' [initial] and 'fair' [new]).

Alongside our whole time-sensor space analysis, we also performed a region-of-interest (ROI) analysis, again *exactly copying the procedure used in previous reports* (*Klinkenberg et al., 2013*; *Caldenhove et al., 2017*). In particular, we focused on the three electrodes Fz, FCz, and Cz. We extracted for every participant the peak MMN amplitude and its latency. We defined the MMN peak as the minimum of the difference wave (ERP to standard tones minus ERP to deviant tones), obtained using the adjusted pre-processing pipeline and the '2REP' trial definition, in the time window between 150 ms and 250 ms. MMN peak amplitudes and latencies per participant and electrode were entered into separate $3 \times 3$ ANOVAs (factor 1: drug group, factor 2: electrode) for each study separately. Significant main effects were followed up by post-hoc *t*-tests.

## Acknowledgements

This study was supported by the University of Zurich (KES) and the René and Susanne Braginsky Foundation (KES). We thank Diana Kutyniok (MPI for Metabolism Research Cologne) for performing DNA isolation and SNP-genotyping.

## Additional information

### Funding

| Funder | Grant reference number | Author |
| --- | --- | --- |
| University of Zurich | | Klaas Enno Stephan |
| René und Susanne Braginsky Stiftung | | Klaas Enno Stephan |
| Max Planck Institute for Metabolism Research | | Klaas Enno Stephan |
| ETH Zürich (open access funding) | | Lilian Aline Weber |

The funders had no role in study design, data collection and interpretation, or the decision to submit the work for publication.

### Author contributions

Lilian Aline Weber, Conceptualization, Data curation, Formal analysis, Investigation, Methodology, Software, Visualization, Writing - original draft, Writing – review and editing; Sara Tomiello, Data curation, Investigation, Writing – review and editing; Dario Schöbi, Katharina V Wellstein, Investigation, Writing – review and editing; Daniel Mueller, Resources, Writing – review and editing; Sandra Iglesias, Conceptualization, Project administration, Supervision, Writing – review and editing; Klaas Enno Stephan, Conceptualization, Funding acquisition, Supervision, Writing – review and editing

### Author ORCIDs

Lilian Aline Weber ⓘ http://orcid.org/0000-0001-9727-9623
Sandra Iglesias ⓘ http://orcid.org/0000-0002-1778-7239
Klaas Enno Stephan ⓘ http://orcid.org/0000-0002-8594-9092

## Ethics

Human subjects: All participants gave written informed consent prior to data acquisition and were financially reimbursed for their participation. The study was approved by the cantonal Ethics Committee of Zurich (KEK-ZH-Nr. 2011-0101/3).

## Decision letter and Author response

Decision letter https://doi.org/10.7554/eLife.74835.sa1
Author response https://doi.org/10.7554/eLife.74835.sa2

---

# Additional files

## Supplementary files

• Supplementary file 1. Significant clusters of activation for a reduced sample size in study 1 (Table S1) and study 2 (Table S2). Tables show the results for the main contrasts reported in the main text when excluding data sets due to lack of behavioral data or low performance in the visual distraction task.

• Transparent reporting form

## Data availability

All raw data (EEG data, behavior) used for this manuscript are available at https://research-collection.ethz.ch/handle/20.500.11850/477685, adhering to the FAIR (Findable, Accessible, Interoperable, and Re-usable) data principles. The analysis code that reproduces the results presented here is publicly available on the GIT repository of ETH Zurich at https://gitlab.ethz.ch/tnu/code/weber-muscarinic-mmn-erp-2021.

The following dataset was generated:

| Author(s) | Year | Dataset title | Dataset URL | Database and Identifier |
|---|---|---|---|---|
| Weber LA, Tomiello S, Schöbi D, Wellstein KW, Müller D, Iglesias S, Stephan KE | 2021 | Auditory mismatch responses are differentially sensitive to changes in muscarinic acetylcholine versus dopamine receptor function | https://doi.org/10.3929/ethz-b-000477685 | ETH Library research collection, 10.3929/ethz-b-000477685 |

---

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

# Appendix 1

## A model-based perspective on trial-by-trial auditory ERPs

We have previously modelled the trial-wise auditory ERPs in MMN paradigms using a hierarchical Bayesian model of belief updating (*Mathys et al., 2011*) and demonstrated that later mismatch responses reflect higher-level prediction errors involved in estimating the stability of the auditory environment (*Weber et al., 2020*). However, volatility effects on MMN *have also previously been demonstrated using a "contextual" approach* (contrasting phases with different stability levels) (*Todd et al., 2014*; *Dzafic et al., 2020*). In the main text, we followed this approach, and experimentally demonstrate an interaction between mismatch and stability, *supporting the pure model-based analyses* (*Weber et al., 2020*) that have operated under *constant volatility paradigms.*

Here, we complement the block-wise model-free analysis by additionally reporting the results of a model-based analysis, which we had fully specified already in our analysis plan. For convenience, we include the paragraphs describing this analysis here again.

### Trial definition
For the single-trial analysis, we only defined one trial type ('tone'), i.e., we kept all tone events, and we did not perform an average over trials.

### Conversion to images and smoothing
We converted the final pre-processed file of each participant into 4D (scalp × within-trial time points × single trials) images using the SPM function spm_eeg_convert2images. Subsequently, we applied a spatial smoothing with a Gaussian kernel of 16 mm FWHM in both spatial (x and y) dimensions. These smoothed images entered a first level regression GLM per participant.

### Model
In the absence of any behavioral responses of participants to the auditory input, we modeled all participants as surprise-minimizing ('Bayes-optimal') agents under the perceptual model of the 3-level hierarchical Gaussian Filter (HGF) for binary inputs. This means that we chose the perceptual parameters of this model such that an exposure to the tone sequence used in our task leads to least surprise over all inputs. To this end, we used the Matlab function tapas_fitModel from the HGF toolbox (v5.1.0), distributed as part of TAPAS (release v3.0.0) with the tapas_bayes_optimal_binary function as a pseudo response model. We previously examined the robustness of this parameter estimation for the given tone sequence with respect to changes in the prior means and uncertainties for each parameter. The results of this analysis are documented in DPRST_HGF_Bayes_optimal_parameters.pdf (available as part of the publicly available analysis plan). As a result, we used the belief trajectories of the agent characterized by the parameter values (and starting values) summarized in the last row of *Appendix 1—table 1*.

From the belief trajectories of this agent, we extracted the trial-wise estimates of precision-weighted prediction error on two levels of the hierarchy, $abs\,(\varepsilon_2)$ and $\varepsilon_3$ , corresponding to PEs about stimulus occurrences and learning signals for the estimation of environmental volatility, respectively. These vectors entered the first level GLM of trial-wise EEG signals for each participant as multiple regressors.

### First-level GLMs
Our model-based vectors of precision-weighted PEs served as regressors in a general linear model (GLM) of trial-wise EEG signals for each participant separately, correcting for multiple comparisons over the entire time-sensor matrix, using Gaussian random field theory. We did not orthogonalize the regressors. We used the same regressors for all participants but excluded entries for trials that were rejected during EEG preprocessing to ensure the regressors have the same length as the number of available EEG data trials for each participant. On the first level, we used a multiple regression model and defined separate *t*-tests for each regressor to examine positive and negative correlations of EEG amplitudes with each regressor. This GLM only served to provide the beta images used for the second level analysis, therefore, we did not threshold the images based on the significance of the tests under either peak- or cluster-level familywise error (FWE) correction.

### Group-level GLMs

The GLMs and tests on the second level for the model-based analysis were specified exactly as the GLMs for the group level conventional ERP analysis: separate GLMs for each computational quantity, which implement a factorial design with the between-subject factor 'Drug group', a covariate for drug plasma levels, and the same second level $t$ contrasts for each computational quantity as in the conventional ERP analysis for each contrast of interest (see main text).

### Visualization as ERPs

To visualize the effects of our model-based analysis in terms of ERPs, we used an additional trial definition which averaged the ERPs for the 10% lowest prediction error trials ('standards') and the 10% highest prediction error trials ('deviants') and computed the grand averages across participants and difference waves for these ERPs.

## Results

Instead of averaging the EEG response to tones assigned to different categories ('standards', 'deviants'), here, we extracted an estimate of precision-weighted prediction error elicited by each tone in the sequence, according to a hierarchical Bayesian model of inference and learning (*Mathys et al., 2011*). This model provides prediction errors on two levels of a belief hierarchy: low-level prediction errors about tone occurrences, which serve to update beliefs about the current tendency in the environment to present one or the other tone, and higher-level prediction errors about tone probabilities, which serve to update an estimate of the current level of environmental volatility. In the following, we report clusters in which trial-wise EEG amplitudes varied in accordance with either of these PEs, and the effects of the pharmacological manipulations on the relationship between EEG and PEs.

### Low-level precision-weighted prediction error

In both studies, and across drug groups, there was a significant relation between $\varepsilon_2$, our model-based trial-wise estimate of low-level precision-weighted PE, and trial-wise EEG activity corresponding to the classical MMN in frontal, fronto-central and central sensors, as well as some later clusters (*Appendix 1—figures 1 and 2*; *Appendix 1—Tables 2 and 3*).

In study 1, we found a significant effect of drug group on the relationship between $\varepsilon_2$ and EEG amplitudes. Compared to the placebo group, ERP amplitudes in the biperiden group showed a weaker positive relation with $\varepsilon_2$ at left central sensors around 330 ms after tone onset (*Appendix 1—figure 1B*; and see *Appendix 1—table 2* for two additional very small clusters). There were no significant differences between drug groups in the effect of $\varepsilon_2$ in study 2.

### High-level precision-weighted prediction error

In both studies, and across drug groups, we found significant trial-by-trial relations between $\varepsilon_3$ (the precision-weighted PE that serves to update volatility estimates) and EEG amplitudes mainly corresponding to a P3-like late central positivity (roughly 250–400ms, peaking around 300ms) and some additional smaller clusters (*Appendix 1—figure 1C* and *Appendix 1—figure 2B*; *Appendix 1—Tables 2 and 3*).

In study 1, we found a significant effect of drug group on the relationship between $\varepsilon_3$ and EEG amplitudes. Compared to the placebo group, trial-wise amplitudes of the late central positivity in the biperiden group showed a stronger relation with $\varepsilon_3$ around 360 ms after tone onset (*Appendix 1—figure 1D*; *Appendix 1—table 2*). There were no significant differences between drug groups in the effect of $\varepsilon_3$ in study 2.

## Conclusions

As in our previously reported analysis of single-trial MMN responses in terms of hierarchical prediction errors (*Weber et al., 2020*), here we find that low-level precision-weighted prediction errors about stimulus occurrences are encoded in trial-wise EEG amplitudes corresponding to the classical mismatch negativity, while higher-level precision-weighted prediction errors, serving as learning signals for an estimation of environmental volatility, correlate with trial-wise EEG amplitudes in a later time window. Our model-based analysis of auditory mismatch locates the dominant pharmacological effect of biperiden in this later time window, similar to the results of our whole time × sensor space analysis with an increase in trial numbers per condition (*Figure 2—figure supplement 3* of the main

text). Interestingly, under biperiden compared to placebo, we find a decrease in the correlation of EEG amplitudes with low-level PEs, but an increase in correlation with high-level (volatility) PEs, reminiscent of the relatively strong interaction effect mismatch*stability under biperiden reported in the main text.

However, when relating the results of our model-based EEG analysis to the more conventional averaging approach we followed in the main text, it is important to note that there is no reason to expect the effects of $\varepsilon_2$ and $\varepsilon_3$ to simply map onto the effects of mismatch and stability (or volatility), respectively. Specifically, the HGF is a model which already takes into account the effects of environmental volatility on belief updates by scaling the current learning rate (i.e., the precision-weight on the PE). Thus, while we did hypothesize $\varepsilon_2$ to capture the classical mismatch negativity component of the auditory ERPs, the $\varepsilon_2$ regressor itself also already scales with current volatility estimates on a trial-by-trial basis, that is, it includes a fine-grained version of what we are trying to capture coarsely in the stability and the interaction effect (mismatch*stability) in the conventional approach.

Moreover, the higher level PE $\varepsilon_3$ does not correspond to the current level of (estimated) volatility – this is captured in the model's variable $\mu_3$ which quantifies an individual's estimate of volatility. Instead, $\varepsilon_3$ is a belief *update* signal, which quantifies the amount to which the estimate of current speed of change in the environment (volatility) should be adjusted based on how far off the agent's belief about the statistical laws driving the stimulus occurrences was.

Importantly, while this quantity is also expected to scale with the more slowly changing volatility beliefs, it is still a function of the trial-by-trial input (e.g., it shows opposite signs in response to expected tones [standards] and unexpected tones [deviants]). Because of this, it is also expected to capture ERP components which differ between standard and deviant trials – that is classical mismatch components, which is exactly what we find. In general, ERPs, due to their transient nature, have been hypothesized to mainly reflect such belief updates. In the specific case of auditory oddball paradigms, belief updates are believed to take place on multiple levels of a processing hierarchy. It is thus an interesting finding – and very consistent with our previous report in a different data set (*Weber et al., 2020*) – that later components of the auditory ERP (such as the P300) seem to reflect belief updates on a higher level, such as volatility estimation.

In summary, our model-based analysis provides a complementary perspective on auditory mismatch signals in our task, but agrees with the results of the conventional categorical standard/deviant approach reported in the main text in the following important points:

1. Biperiden affects auditory mismatch signals, while we find no evidence for a dopaminergic modulation of MMN by amisulpride or levodopa.

2. Biperiden does not only affect mismatch detection itself, but also the higher-level learning about the volatility of the environment – indicated by a stronger interaction effect mismatch*stability in the biperiden group (main text), as well as a significant effect of biperiden on the representation of the higher-level PE $\varepsilon_3$ which serves to update volatility beliefs.

3. Galantamine, at the dose given here, did not affect mismatch processing (or its scaling by environmental stability) in our paradigm.

**Appendix 1—table 1.** Parameter settings for the HGF.
The first two rows show the priors under which the surprise-minimizing parameters were estimated, the last row displays the result of the estimation and thus the values used in the simulation of belief trajectories. All perceptual parameters and starting values of beliefs were fixed to their prior means (indicated by zero prior variance) except for the tonic learning rate on the second level, $\omega 2$.

| Parameter | $\mu_2^{(0)}$ | $\mu_3^{(0)}$ | $\sigma_2^{(0)}$ | $\sigma_3^{(0)}$ | $\kappa_1$ | $\kappa_2$ | $\omega_2$ | $\omega_3$ |
|---|---|---|---|---|---|---|---|---|
| Prior mean | 0 | 1 | 0.25 | 1 | 1 | 1 | -3 | −10 |
| Prior variance | 0 | 0 | 0 | 0 | 0 | 0 | 16 | 0 |
| Posterior mean | 0 | 1 | 0.25 | 1 | 1 | 1 | −3.03 | −10 |

**Appendix 1—table 2.** Results of the model-based single-trial analysis in study 1.
Columns are organized as in *Table 1* of the main text.

| Study 1: Model | cluster | $x$ [mm] | $y$ [mm] | $z$ [ms] | $t_{66}$ | $Z_{\equiv}$ | $p_{FWE}$ | $k_E$ | $tw_{sig}$[ms] |
|---|---|---|---|---|---|---|---|---|---|
| Low-level PE: pos. | 1 | –60 | –57 | 176 | 14.19 | Inf | 0.000 | 5583 | 108–228 |
| | | –17 | 72 | 144 | 12.43 | Inf | 0.000 | | |
| | | –8 | 72 | 144 | 12.41 | Inf | 0.000 | | |
| | 2 | 47 | –36 | 360 | 6.53 | 5.71 | 0.000 | 1353 | 284–400 |
| | | 42 | –41 | 336 | 6.51 | 5.70 | 0.000 | | |
| | | 47 | –41 | 400 | 5.71 | 5.13 | 0.001 | | |
| | 3 | –55 | –25 | 400 | 6.06 | 5.38 | 0.000 | 423 | 360–400 |
| | 4 | 68 | 18 | 192 | 4.79 | 4.42 | 0.013 | 4 | 184–196 |
| | 5 | –42 | –30 | 312 | 4.58 | 4.25 | 0.025 | 12 | 312–316 |
| Low-level PE: neg. | 1 | -8 | -3 | 176 | 16.40 | Inf | 0.000 | 7361 | 100–232 |
| | | 4 | 29 | 168 | 15.84 | Inf | 0.000 | | |
| | 2 | –13 | –14 | 400 | 6.51 | 5.70 | 0.000 | 663 | 364–400 |
| Low-level PE: PLA >BIP | 1 | –26 | –30 | 328 | 4.46 | 4.15 | 0.036 | 8 | 328–332 |
| Low-level PE: BIP > PLA | 1 | 42 | –62 | 336 | 4.91 | 4.51 | 0.009 | 47 | 328–344 |
| | 2 | 38 | –62 | 392 | 4.43 | 4.13 | 0.040 | 10 | 388–396 |
| High-level PE: pos. | 1 | -8 | –36 | 324 | 8.48 | 6.95 | 0.000 | 3679 | 248–400 |
| | | –13 | –46 | 364 | 7.77 | 6.52 | 0.000 | | |
| | | –26 | –68 | 400 | 5.60 | 5.04 | 0.001 | | |
| | 2 | 4 | 72 | 392 | 6.53 | 5.71 | 0.000 | 208 | 364–400 |
| | 3 | 8 | –95 | 220 | 5.21 | 4.75 | 0.003 | 79 | 212–228 |
| | 4 | 4 | 61 | 212 | 4.46 | 4.15 | 0.036 | 11 | 208–212 |
| | | 21 | 61 | 212 | 4.41 | 4.11 | 0.042 | | |
| High-level PE: neg. | 1 | 64 | –52 | 328 | 6.79 | 5.89 | 0.000 | 2483 | 272–400 |
| | | 34 | 13 | 380 | 6.10 | 5.41 | 0.000 | | |
| | | 60 | –36 | 284 | 6.06 | 5.38 | 0.000 | | |
| | 2 | –55 | –46 | 308 | 4.85 | 4.47 | 0.011 | 165 | 296–320 |
| | 3 | 0 | 50 | 276 | 4.77 | 4.40 | 0.014 | 20 | 260–280 |
| | 4 | 30 | –30 | 216 | 4.76 | 4.40 | 0.014 | 97 | 208–224 |
| | | 4 | 2 | 216 | 4.59 | 4.26 | 0.024 | | |
| | 5 | 13 | –95 | 320 | 4.49 | 4.18 | 0.033 | 4 | 316–324 |
| | 6 | –21 | 40 | 256 | 4.41 | 4.11 | 0.042 | 2 | 252–256 |
| High-level PE: BIP > PLA | 1 | -8 | –14 | 360 | 5.35 | 4.86 | 0.002 | 327 | 336–376 |

**Appendix 1—table 3.** Results of the model-based single-trial analysis in study 2. Columns are organized as in *Table 1* of the main text.

| Study 2: Model | cluster | $x$ [mm] | $y$ [mm] | $z$ [ms] | $t_{73}$ | $Z_{\equiv}$ | $p_{FWE}$ | $k_E$ | $tw_{sig}$[ms] |
|---|---|---|---|---|---|---|---|---|---|
| Low-level PE: pos. | 1 | –47 | –68 | 168 | 14.03 | Inf | 0.000 | 6196 | 100–224 |
| | | 0 | 72 | 164 | 13.51 | Inf | 0.000 | | |
| | | 47 | –73 | 192 | 12.61 | Inf | 0.000 | | |
| | 2 | –34 | –68 | 400 | 7.21 | 6.24 | 0.000 | 432 | 368–400 |

*Appendix 1—table 3 Continued on next page*

*Appendix 1—table 3 Continued*

| Study 2: Model | cluster | $x\,[mm]$ | $y\,[mm]$ | $z\,[ms]$ | $t_{73}$ | $Z_{\equiv}$ | $p_{FWE}$ | $k_E$ | $tw_{sig}[ms]$ |
|---|---|---|---|---|---|---|---|---|---|
| | 3 | −17 | −25 | 256 | 6.85 | 6.00 | 0.000 | 1386 | 240–300 |
| | 4 | 0 | 72 | 400 | 6.51 | 5.76 | 0.000 | 225 | 364–400 |
| | 5 | 38 | −73 | 400 | 6.31 | 5.62 | 0.000 | 147 | 384–400 |
| | 6 | 68 | 18 | 176 | 4.51 | 4.22 | 0.032 | 5 | 168–184 |
| Low-level PE: neg. | 1 | 21 | 8 | 156 | 17.50 | Inf | 0.000 | 8067 | 100–220 |
| | | 13 | 8 | 176 | 17.07 | Inf | 0.000 | | |
| | | 34 | −3 | 124 | 11.88 | Inf | 0.000 | | |
| | 2 | -8 | −14 | 400 | 9.01 | 7.37 | 0.000 | 1582 | 360–400 |
| | 3 | −42 | −73 | 264 | 5.44 | 4.97 | 0.002 | 221 | 252–296 |
| | | −26 | −89 | 264 | 5.29 | 4.85 | 0.003 | | |
| | | −60 | −57 | 268 | 5.27 | 4.84 | 0.003 | | |
| | 4 | 42 | 50 | 324 | 5.34 | 4.89 | 0.002 | 74 | 296–332 |
| | | 34 | 61 | 300 | 4.74 | 4.41 | 0.016 | | |
| | | 60 | 29 | 324 | 4.68 | 4.36 | 0.019 | | |
| | 5 | 64 | −62 | 284 | 5.01 | 4.63 | 0.007 | 19 | 280–292 |
| | 6 | 34 | 61 | 264 | 4.40 | 4.13 | 0.044 | 2 | 264–264 |
| High-level PE: pos. | 1 | 0 | −25 | 304 | 6.57 | 5.80 | 0.000 | 619 | 260–320 |
| | | 0 | −25 | 272 | 5.38 | 4.92 | 0.002 | | |
| | 2 | -4 | −68 | 148 | 4.90 | 4.54 | 0.010 | 41 | 140–156 |
| High-level PE: neg. | 1 | −42 | −73 | 304 | 5.66 | 5.13 | 0.001 | 111 | 288–316 |
| | 2 | -4 | 56 | 268 | 5.26 | 4.82 | 0.003 | 121 | 248–296 |
| | 3 | 42 | −78 | 104 | 4.52 | 4.23 | 0.034 | 6 | 104–108 |
| | 4 | −60 | −57 | 308 | 4.48 | 4.20 | 0.038 | 2 | 304–308 |

## Model-based perspective (study 1)

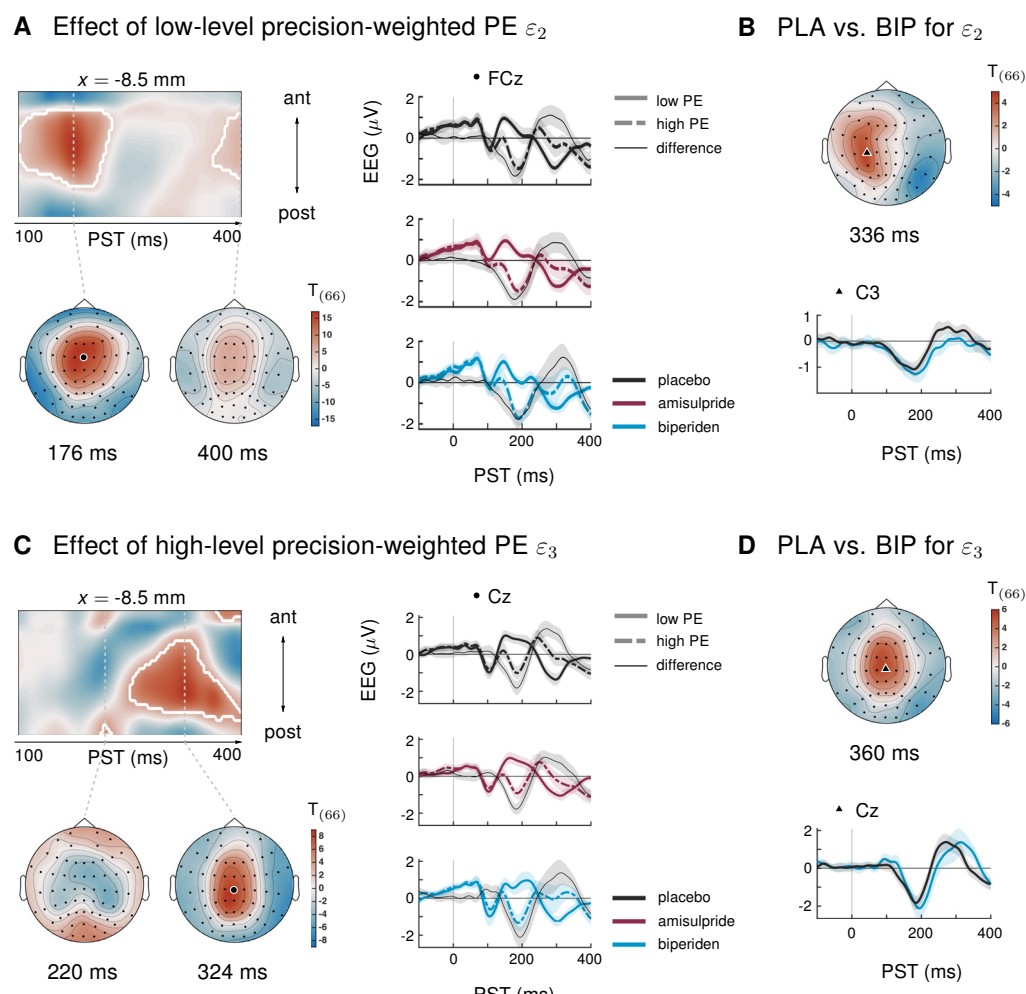

**Appendix 1—figure 1.** Results of the model-based single-trial analysis in study 1. (**A**) The effect of the lower-level precision-weighted PE $\varepsilon_2$ on EEG amplitudes. Left: Regions of the time × sensor space where ERPs to tones were significantly modulated by $\varepsilon_2$. Logic of display as in *Figure 2* in the main text. Right: ERPs and difference waves for selected sensors, separately for the three drug groups. (**B**) Pharmacological effects on $\varepsilon_2$: we found a weaker modulation of ERPs by $\varepsilon_2$ under biperiden in left central sensors. Lower plot shows the ERP difference waves (ERPs to high PE tones – ERPs to low PE tones). (**C**) The effect of the higher level precision-weighted PE $\varepsilon_3$ on EEG amplitudes. (**D**) Biperiden shifted the late central positivity encoding the higher level PE in time.

## Model-based perspective (study 2)

**A** Effect of low-level precision-weighted PE $\varepsilon_2$

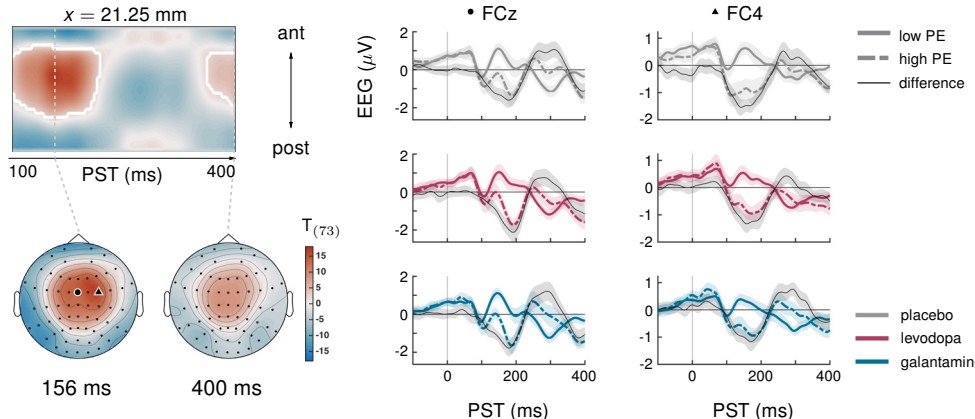

**B** Effect of high-level precision-weighted PE $\varepsilon_3$

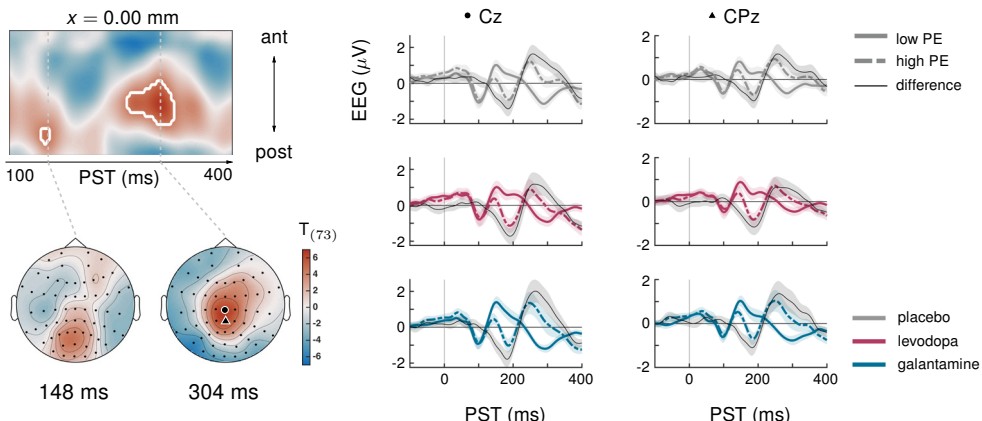

**Appendix 1—figure 2.** Results of the model-based single-trial analysis in study 2. (**A**) The effect of the lower-level precision-weighted PE $\varepsilon_2$ on EEG amplitudes. Left: Regions of the time × sensor space where ERPs to tones were significantly modulated by $\varepsilon_2$. Logic of display as in **Figure 2** in the main text. Right: ERPs and difference waves for selected sensors, separately for the three drug groups. (**B**) The effect of the higher-level precision-weighted PE $\varepsilon_3$ on EEG amplitudes. There were no significant differences between drug groups in either prediction error.

## Appendix 2

### Main effect of stability and interaction stability ×drug group

#### Study 1: Main effect of stability

ERPs in three clusters showed a significant main effect of stability (when averaging over standard and deviant tones and the three drug groups): early on, pre-frontal (144–148 ms, peak at 148 ms, $t$ = 4.76, p = 0.017) and occipital sensors (136–156 ms, peak at 144 ms, $t$ = 5.04, p = 0.007) showed stability effects; later on, stability affected ERPs at parietal sensors (272–292 ms, peak at 284 ms, $t$ = 5.32, p = 0.003; see *Appendix 2—table 1* and *Appendix 2—figure 1*). In all three clusters, ERPs to tones in stable phases were significantly more positive than ERPs to tones in volatile phases (see *Appendix 2—figure 1B*).

#### Study 1: Stability × drug group interaction

While the late effect of stability was quite consistent across drug groups, the earlier effects, particularly at pre-frontal channels, were expressed most prominently in the placebo group. In contrast, the ERPs in the biperiden group displayed a different effect early on: responses to both standard and deviant tones around 124 ms at central sensors were less negative during volatile phases than during stable phases. In other words, the N1 component of the auditory ERP to tones was less pronounced in volatile phases than during stable phases, and this N1 decrement with volatility was not observed in the other groups.

However, when directly contrasting the impact of stability on the auditory ERPs between drug groups, we found no significant differences, both across the whole time × sensor space and within the functionally defined mask.

#### Study 2: Main effect of stability

In study 2, ERPs to tones in volatile phases were significantly more positive than ERPs to tones in stable phases between 268 ms and 272 ms at frontal sensors (peak at 268 ms, $t$ = 4.22, p = 0.035; and at 384 ms, $t$ = 4.42, p = 0.046; *Appendix 2—table 1*, *Appendix 2—figure 2*). Plotting stable and volatile ERPs at these sensors separately for the three drug groups shows a sustained difference between the conditions in the galantamine group, whereas the effects appear to be more transient in the other groups (*Appendix 2—figure 2B*).

#### Study 2: Stability × drug group interaction

The impact of stability on auditory ERPs differed significantly between the drug groups in two places. Between 200 ms and 208 ms, left prefrontal sensors showed significantly more positive modulation by stability (ERP amplitudes being more positive for tones in stable phases) in the galantamine group compared to the placebo group (peak at 204 ms, $t$ = 4.73, p = 0.018). At 264 ms, the levodopa group differed from the placebo group ($t$ = 4.58, p = 0.028) in that left central and fronto-central sensors showed more positive amplitudes in response to tones in volatile compared to stable phases, while the opposite effect was visible in the placebo group (*Appendix 2—figure 2C*). There were no additional differences between drug groups when constraining the search volume using the average effect of stability.

#### Effects of stability: conclusions

In summary, we also found that stability itself modulated auditory ERPs, irrespective of predictability (standards vs. deviants), in both studies, although these effects appear small and only partly replicate across the two studies. The most consistent finding is an effect on later ERP components around 272 ms, which manifested as a positive modulation (stable > volatile) at parietal sensors (study 1), and a negative modulation (volatile > stable) at frontal sensors (study 2). In study 2, this effect was strongly driven by the galantamine group. While an effect of galantamine on the modulation of evoked responses by stability/volatility would be intriguing, the observed effects in the present study appear as ERP-wide differences rather than modulations of specific ERP components. Our pre-processing choices (weak high-pass filtering, no baseline correction) do not exclude the possibility that slow drifts were affecting the offset of the grand averages in the stable and volatile conditions differentially. Unlike deviance, stability was manipulated in blocks rather than on a trial-by-trial basis. This might have led to the apparent ERP differences in the galantamine group.

Importantly, interaction effects between deviance and stability, as we report them in the main text, do not suffer from this constraint: the quickly alternating standard and deviant trials would be affected equally by tonic offsets, thus, any interaction of the stability effects with mismatch (standards vs. deviants) cannot be explained by the impact of artifactual slow drifts.

**Appendix 2—table 1.** Significant clusters for main effects of stability and interactions stability × drug group across the two studies.

Columns are organized as in *Table 1* of the main text.

| Effects of stability | cluster | $x$ [mm] | $y$ [mm] | $z$ [ms] | $t_{66/73}$ | $Z_{\equiv}$ | $p_{FWE}$ | $k_E$ | $tw_{sig}$[ms] |
|---|---|---|---|---|---|---|---|---|---|
| **A** study 1: stable >volatile | 1 | 4 | –62 | 284 | 5.32 | 4.83 | 0.003 | 110 | 272–292 |
| | 2 | 4 | –95 | 144 | 5.04 | 4.62 | 0.007 | 45 | 136–156 |
| | 3 | 0 | 61 | 148 | 4.76 | 4.39 | 0.017 | 14 | 144–148 |
| **B** study 2: volatile >stable | 1 | 0 | 40 | 268 | 4.51 | 4.22 | 0.035 | 6 | 268–272 |
| | 2 | 4 | 24 | 384 | 4.42 | 4.15 | 0.046 | 4 | 384–384 |
| **C** study 2: PLA >LEV | 1 | –38 | –14 | 264 | 4.58 | 4.28 | 0.028 | 4 | 264–264 |
| **D** study 2: GAL > PLA | 1 | –30 | 61 | 204 | 4.73 | 4.40 | 0.018 | 7 | 200–208 |

## Effects of stability (study 1)

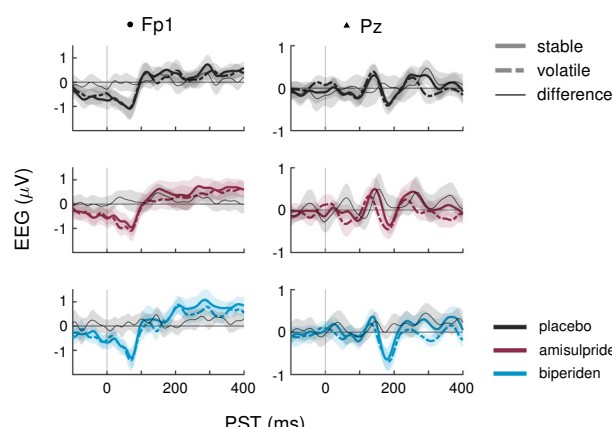

**Appendix 2—figure 1.** Main effect of stability in study 1. (**A**) Regions of the time ×sensor space where ERPs to tones in stable phases were more positive than ERPs to tones in volatile phases. Logic of display as in *Figure 2* in the main text. We found early (at 148ms in pre-frontal sensors and at 144ms in occipital sensors) and late effects (at 284ms in parietal sensors) of stability on the ERPs. (**B**) ERPs and difference waves for selected sensors, separately for the three drug groups. In all clusters, ERPs to tones in stable phases were more positive than ERPs to tones in volatile phases.

## Effects of stability (study 2)

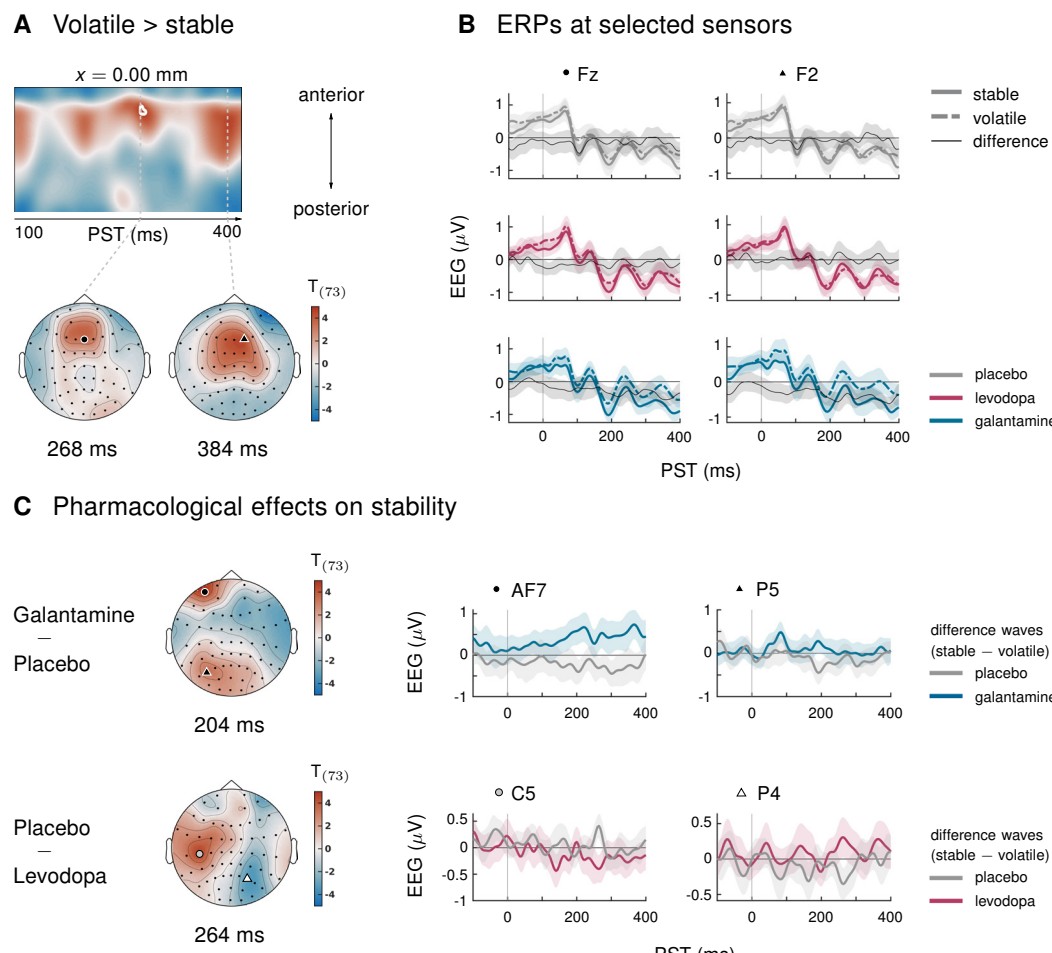

**Appendix 2—figure 2.** Main effect of stability and interaction stability ×drug group in study 2. (**A**) Regions of the time ×sensor space where ERPs to tones in volatile phases were more positive than ERPs to tones in stable phases. Logic of display as in *Figure 2* in the main text. (**B**) ERPs and difference waves for selected sensors, separately for the three drug groups. In this cluster, ERPs to tones in volatile phases were more positive than ERPs to tones in stable phases. This effect was driven by the galantamine group. (**C**) Pharmacological effects on stability: we found significant differences in the impact of stability on ERPs in two clusters. Plots show the ERP difference waves (ERPs to tones in stable phases − ERPs to tones in volatile phases) in different drug conditions.

## Appendix 3

### Blood analysis for genetic variation

We collected an additional blood sample per participant to assess genetic variation at functional single nucleotide polymorphisms (SNPs) of two genes relevant to the pharmacological intervention. For genetic analyses, 20 ml of blood were collected in tubes containing ethylene-diaminetetraacetic as anticoagulant, centrifuged at 10 °C for 10 min at 3000xg to separate the buffy coat and finally stored at –86 °C until analysis. In this study, we assessed genetic variations at SNPs of the genes that code for the Catechol-O-methyl-transferase (COMT, using the rs4680 SNP) and the choline acetyltransferase (ChAT, using the rs1880676 SNP) enzymes.

Specifically, DNA from the buffy coat was isolated using the QIAamp DNA Blood Mini Kit (Cat No. 51106, Qiagen GmbH, Hilden, Germany) according to manufacturer's protocol. Concentration and quality of the DNA was assessed with a UV/Vis-spectrophotometer (ND-1000, Peqlab GmbH, Erlangen, Germany). Then, 20 ng of DNA was analyzed in triplicates using allelic discrimination assays (TaqMan SNP Genotyping Assays, Applied Biosystems by Thermo Fisher Scientific Inc, Waltham, MA, USA). Genotyping PCR was performed on a 7900HT Fast Real-Time PCR system (Applied Biosystems) and the data analyzed with Sequence Detection Software (SDS) 2.3 (Applied Biosystems). DNA isolation and SNP-genotyping was performed by the Max Planck Institute for Metabolism Research in Cologne. These procedures were equivalent for both studies.

### GLM specification for genetic effects

In two additional GLMs per effect of interest, we considered variations in the expression of our experimental factors in EEG activity that are due to different variants (SNPs) of two genes that determine the availability of dopamine (DA) and ACh in the brain, respectively: COMT – which encodes catechol-O-methyltransferase, a key enzyme for the degradation of DA – and CHAT – which encodes choline acetyltransferase (abbreviated as ChAT), the enzyme responsible for synthesis of ACh. In one GLM, an additional covariate 'COMT polymorphism' coded for SNPs of the COMT gene, in the other GLM, an equivalent covariate 'CHAT polymorphism' was used. These covariates served to account for the possible contribution of genetic effects to individual variability in DA/ACh availability. They were used to test for direct effects on mismatch-related EEG activations and for pharmaco-genetic interaction effects on brain activity during mismatch processing. In specifying these covariates, we allowed for an interaction with the drug factor, and mean-centered the covariates within drug groups.

We examined whether these covariates explained additional variance using *t*-tests for positive and negative effects of the covariates (for COMT: within placebo and amisulpride groups, for CHAT: within placebo and biperiden groups) and tested for pharmaco-genetic interaction effects using differential contrasts (COMT: placebo vs. amisulpride; CHAT: placebo vs. biperiden) in both directions. Again, we performed this analysis once across the whole time-sensor space, and subsequently investigated genetic effects within a smaller, functionally constrained search volume of significant average effects (orthogonal contrast). The functional masks were identical to the ones described for the main group-level GLMs.

In study 2, genotyping was inconclusive for two individuals for both gene variants (N = 1: levodopa, N = 1: galantamine), and additionally for one individual only for the COMT gene (N = 1: levodopa). Therefore, the GLMs for CHAT effects in the EEG are based on a sample of N = 76, and those for COMT effects are based on a sample of N = 75.

### Results: genetic effects and pharmaco-genetic interactions

To examine effects of individual differences in dopamine metabolism on mismatch related ERPs and potential interactions of these differences with the effects of amisulpride, we added a covariate coding for genetic variations (SNPs) in the COMT polymorphism to the group level GLMs. In study 1, we found a significant modulation of mismatch ERPs by COMT SNP in the amisulpride group in parieto-occipital sensors (Val/Val > Val/Met > Met/Met, 300–308 ms, peak at 304 ms, *t* = 4.51, p = 0.033, **Appendix 3—table 1**), and in right central sensors (Met/Met > Val/Met > Val/Val, 292–304 ms, peak at 300 ms, *t* = 4.79, p = 0.014, **Appendix 3—table 1**).

Similarly, in separate GLMs, we considered effects of the CHAT polymorphism on mismatch related ERPs in the placebo and the biperiden group and potential interactions of CHAT SNPs with the effect of biperiden. Here, we found a significant modulation of mismatch ERPs in volatile periods

of the experiment by CHAT polymorphism. At left centro-parietal sensors at 384 ms after tone onset, volatile mismatch effects depended on CHAT SNP within the placebo group (A/A > A/G > G/G, t = 4.57, p = 0.032, *Appendix 3—table 1*) and this modulation was significantly different from the effect of CHAT in the biperiden group at these sensors (significant interaction between CHAT covariate and drug group, 356–492 ms, t = 5.77, p = 0.001, *Appendix 3—table 1*).

In study 2, we found that the amplitude of mismatch ERPs in volatile periods of the experiment depended both on CHAT and COMT polymorphism, but only in the placebo group (see *Appendix 3—table 2* for details).

However, all these results (modulation of mismatch ERPs by COMT in the amisulpride group and differential modulation of volatile mismatch ERPs by CHAT in the placebo compared to the biperiden group, modulation of volatile mismatch ERPs by both CHAT and COMT in the placebo group of study 2) have to be treated with great caution, given that our sample size is very small for detecting (the presumably subtle) genetic effects. For example, in study 1, for COMT, there were only 3 individuals showing the Val/Val genotype in the amisulpride group, and for CHAT, there were only 2 individuals showing the A/A genotype in the biperiden group, and 3 individuals with this genotype in the placebo group. Moreover, none of the genetic effects in the placebo groups replicated across the two studies. Given these limitations, we refrain from interpreting or discussing these genetic effects any further and report them here only for completeness and transparency, and as potential guidance for future follow-up studies with larger sample sizes.

**Appendix 3—table 1.** Genetic effects and pharmaco-genetic interactions in study 1.
Columns are organized as in *Table 1* of the main text. Mismatch ERPs in volatile periods were differentially modulated by CHAT polymorphism in the placebo versus the biperiden group, whereas mismatch ERPs overall and in stable periods were modulated by COMT polymorphism in the amisulpride group.

| Study 1: Genetic effects | cluster | $x$ [mm] | [mm] | [ms] | $t_{63}$ | $Z_{\equiv}$ | $p_{FWE}$ | $k_E$ | $tw_{sig}$[ms] |
|---|---|---|---|---|---|---|---|---|---|
| **A** volatile mismatch: CHAT: PLA >BIP | 1 | −34 | −52 | 380 | 5.77 | 5.15 | 0.001 | 302 | 356–392 |
| **B** volatile mismatch: PLA: pos. effect of CHAT | 1 | −34 | −46 | 384 | 4.57 | 4.23 | 0.032 | 6 | 380–384 |
| **C** volatile mismatch: BIP: neg. effect of CHAT | 1 | −51 | −25 | 384 | 4.49 | 4.17 | 0.040 | 6 | 380–384 |
| **D** mismatch: AMI: pos. effect of COMT | 1 | 0 | −73 | 304 | 4.51 | 4.18 | 0.033 | 18 | 300–308 |
| **E** mismatch: AMI: neg. effect of COMT | 1 | 42 | −19 | 300 | 4.79 | 4.41 | 0.014 | 33 | 292–304 |
| **F** stable mismatch: AMI: neg. effect of COMT | 1 | 42 | −14 | 300 | 4.44 | 4.12 | 0.042 | 6 | 300–304 |

**Appendix 3—table 2.** Genetic effects and pharmaco-genetic interactions in study 2.
Columns are organized as in *Table 1* of the main text. Volatile mismatch ERPs in the placebo group were positively modulated by CHAT polymorphism, and negatively modulated by COMT polymorphism.

| Study 2: Genetic effects | cluster | $x$ [mm] | [mm] | [ms] | $t_{68/67}$ | $Z_{\equiv}$ | $p_{FWE}$ | $k_E$ | $tw_{sig}$[ms] |
|---|---|---|---|---|---|---|---|---|---|
| **A** volatile mismatch: PLA: pos. effect of CHAT | 1 | −13 | −78 | 104 | 4.42 | 4.13 | 0.046 | 4 | 104–108 |
| **B** volatile mismatch: PLA: neg. effect of COMT | 1 | 4 | 45 | 192 | 5.31 | 4.84 | 0.003 | 53 | 180–200 |

