## [Editor Report]

This study adds to the considerable, but often conflicting, work on how neurotransmitter systems contribute to auditory processing dysfunction. The paper details a thorough and careful analysis of an important hypothesis from the point of view of schizophrenia research: do muscarinic and dopaminergic receptors contribute to mismatch negativity effects? The answers could be useful for future treatment allocation in psychosis. The analysis was pre-registered and departures from the planned analysis were well-motivated and clearly described.

---

## [Decision Letter]

**Decision letter after peer review:**

Thank you for submitting your article "Auditory mismatch responses are differentially sensitive to changes in muscarinic acetylcholine versus dopamine receptor function" for consideration by *eLife*. Your article has been reviewed by 2 peer reviewers, and the evaluation has been overseen by Jonas Obleser as the Reviewing Editor and Christian Büchel as the Senior Editor. The reviewers have opted to remain anonymous.

Essential revisions:

1) We think both reviewers give clear, reasonable, and detailed queries below. Please do follow them and respond to them as far as possible. It is likely that we would consult both reviewers again prior to any decision on a revised manuscript.

Further to what is listed as major concerns by both reviewers below,

(2a) A major concern that arose in the editorial-reviewer discussions is the following:

The biperiden effects are either non-significant vs placebo or just at the threshold of significance (SVC) in samples of ~25 per group, so we deem it unlikely that these effects could be very informative at the individual level. Of course this might be due to heterogeneity (as the authors suggest in the Clinical Implications) – but equally, the possibility remains that there might be just weak effects and a noisy measure.

We ask the authors to be clearer about the presentation of these results, and give a more realistic assessment of how/whether this paradigm would be able to assess muscarinic function at the individual level. E.g. have they looked at reliability of the effects at the individual level? We are not insisting on doing that analysis but the authors should spell out what needs to be established and how.

(2b) Note also that *eLife* is now giving the dedicated opportunity of a "Ideas and Speculation" section, so we invite you to be particularly clear in separating what can be said based on the data and what your ideas on potential individual differences are in such a section. ( https://elifesciences.org/inside-*eLife*/e3e52a93/*eLife*-latest-including-ideas-and-speculation-in-*eLife*-papers )

*Reviewer #1 (Recommendations for the authors):*

1) It is not clear if the number of standards and deviants were identical across the initial analysis. On P6 it states that "…only considered deviants presented after 5 repetitions…", 5 repetitions of what? A clearer description of what deviant and standard were used in the analysis would be helpful.

2) Was there a statistical test done for the latency of the mismatch effects across the three drug groups shown in Figure 2D, with the initial pre-processing method? I know using the ROI analysis following the different pre-processing method statistical testing showed a significant latency effect (Table 3).

3) Using baseline correction and/or using more trials per condition revealed a previously hidden interaction effect of stability on MMN (P19 bottom paragraph and Figure 3—supplement 1) for study 2. Was this just found for the placebo group or also for the other drug groups (levodopa and galantamine) as only the effect for the placebo group is shown in Figure 3—supplement 1. The statement on P27 in the Discussion "…we did see these interaction effects more robustly across all drug groups in both studies." should be changed if this was not the case, or data from the levodopa and galantamine groups shown.

4) Is the paradigm used in this study considered a "roving "auditory oddball paradigm? There is confusion regarding this as on P26 the current paradigm is contrasted with the "roving" oddball paradigm used in Moran et al., 2013, however in the methods on P41 and in figure 3—supplement 1 the term roving is used in reference to the current study's paradigm.

*Reviewer #2 (Recommendations for the authors):*

No suggestions for further analysis.

---

## [Author Response]

Essential revisions:(1) We think both reviewers give clear, reasonable, and detailed queries below. Please do follow them and respond to them as far as possible. It is likely that we would consult both reviewers again prior to any decision on a revised manuscript.Further to what is listed as major concerns by both reviewers below.(2a) A major concern that arose in the editorial-reviewer discussions is the following:The biperiden effects are either non-significant vs placebo or just at the threshold of significance (SVC) in samples of ~25 per group, so we deem it unlikely that these effects could be very informative at the individual level. Of course this might be due to heterogeneity (as the authors suggest in the Clinical Implications) – but equally, the possibility remains that there might be just weak effects and a noisy measure.

Thank you for this comment. You are right that the difference between the biperiden and placebo groups appears somewhat weak in our main/initial analysis. However, it shows more strongly in the following analyses:

1. Within our initial processing pipeline, the biperiden vs placebo effects do survive multiple comparison correction across the whole time × sensor space when the analysis is restricted to the stable periods of the experiment – an analysis that we had specified a priori in our analysis plan (Table 5; now Figure 2). This finding is remarkable given that the restriction to stable periods reduces the number of trials per condition available for the comparison. The weakness of the drug effect across the whole sequence might thus be due to the averaging of stable and volatile conditions, which show the effect to different degrees.

2. When adopting an analysis approach aimed at maximising the sensitivity at fronto-central sensors (i.e., the alternative pre-processing pipeline with linked mastoid reference, stronger filtering, and more trials retained in the analysis), mismatch signals in the biperiden group are significantly different from both placebo and amisulpride, again corrected at the whole-volume level (Figure 2—figure supplement 3).

3. When adopting a model-based approach, which takes into account all trials of the experiment, we also find whole-volume corrected significant differences between biperiden and placebo, even under the original pre-processing pipeline (Appendix 1).

In summary, we think that these observations suggest that our predetermined main analysis approach might not have provided the optimal sensitivity for both the drug and the volatility effects, which turned out to be more subtle than we had expected. We now discuss this possibility more explicitly on P.32 and provide suggestions for future analysis strategies for our paradigm:

“In this study, we employed a conventional ERP analysis, but considered all sensors and time points under multiple comparison correction, to detect effects of experimental conditions that manifest as differences in evoked response amplitudes within our time-window of interest. In our main analysis, the effect of biperiden on mismatch signals as compared to placebo appeared relatively subtle but was retrieved with higher effect sizes and survived multiple comparison correction across the whole time × sensor space when (a) focusing on the stable phases of our experiment (in line with our analysis plan, Figure 2D), despite the reduction in trial numbers this restriction entails, (b) using an alternative pre-processing pipeline with more aggressive correction of slow drifts and retaining more trials per condition (Figure 2—figure supplement 3), or (c) adopting a model-based approach which takes into account all trials of the experiment (Appendix 1). Similarly, the effect of environmental volatility showed up more robustly across both studies when either using the alternative pre-processing procedure, or retaining more trials per condition, or both (Figure 3—figure supplement 1). This suggests that the pre-processing and statistical strategy of the main analysis does not have optimal sensitivity for detecting these effects. Thus, for future studies using our paradigm, either focusing the analysis on an ROI based on the average effects presented here, or adopting the model-based analysis approach, would be promising strategies – in particular, when extracting effects in individual participants or patients. Furthermore, our pattern of results – an apparent biperiden-induced shift in mismatch responses from an early to a later peak, and from frontal to central channels – suggests that methods which go beyond the amplitude-based approach used here and exploit the rich temporal information in the EEG signal could help us to further understand the impact of cholinergic neurotransmission on perceptual inference in our task.”

We ask the authors to be clearer about the presentation of these results, and give a more realistic assessment of how/whether this paradigm would be able to assess muscarinic function at the individual level. E.g. have they looked at reliability of the effects at the individual level?We are not insisting on doing that analysis but the authors should spell out what needs to be established and how.

Thank you for this comment. We agree that this aspect has not received enough attention in our previous discussion. Subject-level reliability is indeed an important topic that is increasingly debated in neuromodeling and computational psychiatry research.

Generally, the test-retest reliability of auditory MMN amplitudes has been examined in a number of studies and was found to be high (Light et al., 2012; Roach et al., 2020; Wang et al., 2021). For the paradigm and the readouts used in the current manuscript specifically, this remains to be established (which we now acknowledge in the discussion, see below). Our between-subjects design is not ideal for answering this question: the only option would be to split each participant’s data into subsets and compare MMN amplitudes between them. However, this would leave us with relatively few trials per partition; moreover, it is questionable whether one would expect MMN to be stable across different parts of our sequence, given the dynamic and context-dependent nature of the belief updating process that underlies this signal (and that we are capturing with the model-based analysis). However, the lead author of this manuscript is currently involved in a dedicated within-subjects test-retest reliability study of MMN *and* its modulation by volatility and we hope to report the results in the near future.

We have amended the relevant paragraph in the discussion accordingly:

“To establish the utility of our paradigm in the clinical context, two things would be important. First, the test-retest reliability of the proposed biomarker. Generally, previous studies have shown promising results for the MMN, with high intra-class correlation coefficients (up to > 0.90) and other stability measures in both healthy and clinical populations (Light et al., 2012; Roach et al., 2020; Wang et al., 2021), consistent with the idea that ERPs tend to be idiosyncratic in their expression, but reliable within individuals (Gaspar et al., 2011). However, a dedicated test-retest reliability study for the specific MMN variant in this article has not yet been conducted. Second, prospective patient studies are needed, which test whether this readout of cholinergic neurotransmission is predictive of treatment success in individual patients.”

(2b) Note also that eLife is now giving the dedicated opportunity of a "Ideas and Speculation" section, so we invite you to be particularly clear in separating what can be said based on the data and what your ideas on potential individual differences are in such a section. ( https://elifesciences.org/inside-eLife/e3e52a93/eLife-latest-including-ideas-and-speculation-in-eLife-papers )

Thank you for pointing out this possibility. For our current manuscript, we felt that splitting the discussion according to the different topical aspects (e.g., biperiden effects on mismatch versus stability effects on mismatch) provides for a more accessible reading experience. However, we agree with the importance of differentiating clearly between which ideas are supported by the data versus which ideas are more speculative in nature. We have therefore adjusted our wording in all relevant places to ensure this distinction is clear to the reader.

P.29: “Speculatively, this comparably high tonic volatility could mean that precision-weights on PEs were already high; this might have prevented any increase in sensory precision afforded by galantamine to be expressed in the mismatch ERPs in our study due to a ceiling effect.”

P.33: “Based on our results, we speculate that reduced MMN in patients might be relatively more indicative of cholinergic versus dopaminergic dysregulation of synaptic plasticity.”

Reviewer #1 (Recommendations for the authors):1) It is not clear if the number of standards and deviants were identical across the initial analysis. On P6 it states that "…only considered deviants presented after 5 repetitions…", 5 repetitions of what? A clearer description of what deviant and standard were used in the analysis would be helpful.

We apologise for not being clear about the trial definition. Deviants are generally defined as those tones that break an established regularity. In our paradigm, the regularity is established locally in time by ensuring that one of the two tones has been occurring more frequently in the recent past (compare Figure 1). However, in selecting deviants in the initial analysis, we followed previous studies in only considering those tones as deviants that follow a train of at least 5 repetitions of the other tone. This is to ensure that the regularity has been sufficiently established (locally) before the rule break occurs. We have added the following specifications to our description (P7):

“Following previous studies (Garrido et al., 2008), we only considered those tones as deviants which followed at least 5 repetitions of the other tone (resulting in N_deviants_=119). Equivalently, we defined standards as the 6^th^ repetition of a tone (N_standards_=106) in order to keep trial numbers comparable across conditions.”

2) Was there a statistical test done for the latency of the mismatch effects across the three drug groups shown in Figure 2D, with the initial pre-processing method? I know using the ROI analysis following the different pre-processing method statistical testing showed a significant latency effect (Table 3).

No, we have not performed the ROI analysis for the data processed according to our initial pipeline. As discussed in response to public comments (2) and (3) (please see above), we used an average reference which is problematic for only examining selected electrodes. In the control analysis, the combination of a linked mastoid reference with an ROI based on fronto-central sensors makes more sense, whereas our main analysis allowed for distributed effects of mismatch. Even in the context of the alternative pre-processing approach, we only performed the analysis for comparability with previous reports – the peak-based extraction of amplitudes and latencies is known to be very susceptible to noise (e.g., (Clayson et al., 2013)). Finally, based on the pattern of results in our main analysis one would choose a different ROI than the fronto-central one used in previous studies and in the alternative pipeline.

The conclusions of our article are based on this main analysis which follows our analysis plan. Even though this analysis did not allow us to directly test for latency effects (which we did not expect at the time of writing the analysis plan), the pattern of results lead us to conclude that biperiden temporally and spatially shifts the mismatch response, even before conducting the additional control analyses (as can be seen in the previous version of our bioRxiv preprint which was written before we had performed these analyses).

The reanalysis of our data (with alternative pre-processing, trial definitions, and ROI analysis) was performed in response to reviewer suggestions from a previous submission to a different journal, and it served to answer very specific questions (1. Do our main conclusions about biperiden/galantamine hold if we look at our data in exactly the same way that previous studies on biperiden/galantamine effects on MMN have done? 2. Is the weakness of stability*mismatch interaction effects in study 1, and the lack thereof in study 2, due to low sensitivity in our current analysis pipeline?).

We have now amended our presentation of the ROI based results in the Results section (P.21) to include the caveats of this analysis and remind the reader that our conclusions will be based on our main analysis:

“A further difference between our analysis approach and previous reports on muscarinic and galantamine effects on MMN in the literature (Caldenhove et al., 2017; Klinkenberg et al., 2013; Moran et al., 2013) is the use of region-of-interest (ROI) analyses. To fully account for any differences in analysis approach between our and previous studies, we therefore additionally performed a region-of-interest (ROI) analysis, focusing on exactly those sensors used in these previous studies, and following their (peak-based) approach for extracting MMN amplitudes and latencies in every participant (Caldenhove et al., 2017; Klinkenberg et al., 2013) (for details, see Materials and methods).

In line with the results obtained under our original pipeline, we found that the MMN peak latency in study 1 was increased under biperiden (mean: 181.9ms, std: 3.4) compared to placebo (mean: 168.8ms, std: 3.2, Table 3). Peak amplitudes were not significantly different between drug groups. This is consistent with a temporal shift of the mismatch response in the early (classical) MMN time window of the kind we describe above. In other words, even though the whole time × sensor space analysis under the new processing pipeline had located the dominant drug effect in a later component, we still found evidence for this early MMN shift when focusing on the classical MMN sensors. Note, however, that the peak-based approach of extracting ERP amplitudes and latencies employed in this ROI analysis is known to be susceptible to noise (e.g., (Clayson et al., 2013)) and that we will base our main conclusions on the whole sensor space analysis presented above.”

3) Using baseline correction and/or using more trials per condition revealed a previously hidden interaction effect of stability on MMN (P19 bottom paragraph and Figure 3—supplement 1) for study 2. Was this just found for the placebo group or also for the other drug groups (levodopa and galantamine) as only the effect for the placebo group is shown in Figure 3—supplement 1. The statement on P27 in the Discussion "…we did see these interaction effects more robustly across all drug groups in both studies." should be changed if this was not the case, or data from the levodopa and galantamine groups shown.

We agree that this sentence was not fully supported by the data shown in Figure 3—figure supplement 1. We did find a significant interaction effect across all drug groups, but this figure is only showing the placebo group ERPs.

As explained above in response to point (3), this analysis had the clear purpose of examining the dependence of this interaction effect on processing choices. We therefore did not re-examine all contrasts of the main analysis, but focused on the interaction mismatch*volatility, averaging across drug groups. Indeed, these interaction effects were significant (on average across drug groups) in all the three alternative pipelines. We found it most natural to visualize this using the data from the placebo group (as to not be distracted by the presence of absence of drug effects).

To fully support the relevant statement in the discussion, we now plot the ERPs after averaging across all three drug groups per study, better reflecting the contrast we computed (see updated Figure 3—figure supplement 1).

4) Is the paradigm used in this study considered a "roving "auditory oddball paradigm? There is confusion regarding this as on P26 the current paradigm is contrasted with the "roving" oddball paradigm used in Moran et al., 2013, however in the methods on P41 and in figure 3—supplement 1 the term roving is used in reference to the current study's paradigm.

We apologise for this confusion and being ambiguous about this term. We do not consider our paradigm to be a roving oddball paradigm. What we referred to on P41 (now P.45) and in the figure is the original trial definition (which tones to consider as standards and deviants) which was named ‘roving’ for historical reasons (this is also what is used in the analysis plan, section trial definition). We agree this is very confusing and have adapted the naming of the trial definitions in the manuscript. We now refer to the trial definition of the main analysis as “INIT” (initial) as compared to “2REP” (new) in the control analyses (see also the updated Figure 3—figure supplement 1). On P.45 we write:

“Second, we used a different standard and deviant definition (that we had already specified as part of our analysis plan): instead of only considering tones as standards and deviants that followed at least 5 repetitions of the same tone (initial, “INIT” MMN definition), we instead considered every tone after at least 2 repetitions. Because this results in many more standard tones than deviant tones, we only used a subset of these standard tones, choosing them such that for a given number of preceding tone repetitions, there were as many standards as there were deviants (new, “2REP” MMN definition; note that in the publicly available analysis plan, these definitions were labelled “roving” (initial) and “fair” (new)).”

References

Behrens, T. E. J., Woolrich, M. W., Walton, M. E., and Rushworth, M. F. S. (2007). Learning the value of information in an uncertain world. *Nat Neurosci*, *10*, 1214–1221. https://doi.org/Doi 10.1038/Nn1954

Bodick, N. C., Offen, W. W., Levey, A. I., Cutler, N. R., Gauthier, S. G., Satlin, A., Shannon, H. E., Tollefson, G. D., Rasmussen, K., Bymaster, F. P., Hurley, D. J., Potter, W. Z., and Paul, S. M. (1997). Effects of Xanomeline, a Selective Muscarinic Receptor Agonist, on Cognitive Function and Behavioral Symptoms in Alzheimer Disease. *Archives of Neurology*, *54*(4), 465–473. https://doi.org/10.1001/archneur.1997.00550160091022

Bymaster, F. P., Calligaro, D. O., Falcone, J. F., Marsh, R. D., Moore, N. A., Tye, N. C., Seeman, P., and Wong, D. T. (1996). Radioreceptor binding profile of the atypical antipsychotic olanzapine. *Neuropsychopharmacology*, *14*(2), 87–96. https://doi.org/10.1016/0893-133X(94)00129-N

Caldenhove, S., Borghans, L. G. J. M., Blokland, A., and Sambeth, A. (2017). Role of acetylcholine and serotonin in novelty processing using an oddball paradigm. *Behavioural Brain Research*, *331*(April), 199–204. https://doi.org/10.1016/j.bbr.2017.05.031

Clayson, P. E., Baldwin, S. A., and Larson, M. J. (2013). How does noise affect amplitude and latency measurement of event-related potentials (ERPs)? A methodological critique and simulation study. *Psychophysiology*, *50*(2), 174–186. https://doi.org/10.1111/psyp.12001

Dzafic, I., Randeniya, R., Harris, C. D., Bammel, M., and Garrido, M. I. (2020). Statistical learning and inference is impaired in the non-clinical continuum of psychosis. *The Journal of Neuroscience*. https://doi.org/10.1523/jneurosci.0315-20.2020

Garrido, M. I., Friston, K. J., Kiebel, S. J., Stephan, K. E., Baldeweg, T., and Kilner, J. M. (2008). The functional anatomy of the MMN: A DCM study of the roving paradigm. *NeuroImage*, *42*(2), 936–944. https://doi.org/10.1016/j.neuroimage.2008.05.018

Gaspar, C. M., Rousselet, G. A., and Pernet, C. R. (2011). Reliability of ERP and single-trial analyses. *NeuroImage*, *58*(2), 620–629. https://doi.org/10.1016/j.neuroimage.2011.06.052

Gibbons, A. S., Scarr, E., Boer, S., Money, T., Jeon, W. J., Felder, C., and Dean, B. (2013). Widespread decreases in cortical muscarinic receptors in a subset of people with schizophrenia. *International Journal of Neuropsychopharmacology*, *16*(1), 37–46. https://doi.org/10.1017/S1461145712000028

Ichikawa, J., Dai, J., O’Laughlin, I. A., Fowler, W. L., and Meltzer, H. Y. (2002). Atypical, but Not Typical, Antipsychotic Drugs Increase Cortical Acetylcholine Release without an Effect in the Nucleus Accumbens or Striatum. *Neuropsychopharmacology*, *26*(3), 325–339. https://doi.org/10.1016/S0893-133X(01)00312-8

Johnson, D. E., Nedza, F. M., Spracklin, D. K., Ward, K. M., Schmidt, A. W., Iredale, P. A., Godek, D. M., and Rollema, H. (2005). The role of muscarinic receptor antagonism in antipsychotic-induced hippocampal acetylcholine release. *European Journal of Pharmacology*, *506*(3), 209–219. https://doi.org/10.1016/j.ejphar.2004.11.015

Klinkenberg, I., Blokland, A., Riedel, W. J., and Sambeth, A. (2013). Cholinergic modulation of auditory processing, sensory gating and novelty detection in human participants. *Psychopharmacology*, *225*(4), 903–921. https://doi.org/10.1007/s00213-012-2872-0

Lieder, F., Daunizeau, J., Garrido, M. I., Friston, K. J., and Stephan, K. E. (2013). Modelling Trial-by-Trial Changes in the Mismatch Negativity. *PLoS Computational Biology*, *9*(2). https://doi.org/10.1371/journal.pcbi.1002911

Light, G. A., Swerdlow, N. R., Rissling, A. J., Radant, A., Sugar, C. A., Sprock, J., Pela, M., Geyer, M. A., and Braff, D. L. (2012). Characterization of Neurophysiologic and Neurocognitive Biomarkers for Use in Genomic and Clinical Outcome Studies of Schizophrenia. *PLOS ONE*, *7*(7), e39434. https://doi.org/10.1371/journal.pone.0039434

Malkoff, A., Weizman, A., Gozes, I., and Rehavi, M. (2008). Decreased M1 muscarinic receptor density in rat amphetamine model of schizophrenia is normalized by clozapine, but not haloperidol. *Journal of Neural Transmission*, *115*(11), 1563–1571. https://doi.org/10.1007/s00702-008-0122-8

Mathys, C. D., Daunizeau, J., Friston, K. J., and Stephan, K. E. (2011). A Bayesian foundation for individual learning under uncertainty. *Frontiers in Human Neuroscience*, *5*(May), 1–20. https://doi.org/Artn 39 Doi 10.3389/Fnhum.2011.00039

Moran, R. J., Campo, P., Symmonds, M., Stephan, K. E., Dolan, R. J., and Friston, K. J. (2013). Free Energy, Precision and Learning: The Role of Cholinergic Neuromodulation. *The Journal of Neuroscience*, *33*(19), 8227–8236. https://doi.org/10.1523/JNEUROSCI.4255-12.2013

Raedler, T. J., Knable, M. B., Jones, D. W., Lafargue, T., Urbina, R. A., Egan, M. F., Pickar, D., and Weinberger, D. R. (2000). in vivo olanzapine occupancy of muscarinic acetylcholine receptors in patients with schizophrenia. *Neuropsychopharmacology*, *23*(1), 56–68. https://doi.org/10.1016/S0893-133X(99)00162-1

Raedler, T. J., Knable, M. B., Jones, D. W., Urbina, R. A., Egan, M. F., and Weinberger, D. R. (2003). Central muscarinic acetylcholine receptor availability in patients treated with clozapine. *Neuropsychopharmacology*, *28*(8), 1531–1537. https://doi.org/10.1038/sj.npp.1300210

Raedler, T. J., Knable, M. B., Jones, D. W., Urbina, R. A., Gorey, J. G., Lee, K. S., Egan, M. F., Coppola, R., and Weinberger, D. R. (2003). in vivo determination of muscarinic acetylcholine receptor availability in schizophrenia. *American Journal of Psychiatry*, *160*(1), 118–127. https://doi.org/10.1176/appi.ajp.160.1.118

Roach, B. J., Hamilton, H. K., Bachman, P., Belger, A., Carrión, R. E., Duncan, E., Johannesen, J., Kenney, J. G., Light, G., Niznikiewicz, M., Addington, J., Bearden, C. E., Owens, E. M., Cadenhead, K. S., Cannon, T. D., Cornblatt, B. A., McGlashan, T. H., Perkins, D. O., Seidman, L., … Mathalon, D. H. (2020). Stability of mismatch negativity event-related potentials in a multisite study. International Journal of Methods in Psychiatric Research, 29(2), e1819. https://doi.org/10.1002/mpr.1819

Scarr, E., Cowie, T. F., Kanellakis, S., Sundram, S., Pantelis, C., and Dean, B. (2009). Decreased cortical muscarinic receptors define a subgroup of subjects with schizophrenia. *Molecular Psychiatry*, *14*(11), 1017–1023. https://doi.org/10.1038/mp.2008.28

Scarr, Elizabeth, Hopper, S., Vos, V., Suk Seo, M., Everall, I. P., Aumann, T. D., Chana, G., and Dean, B. (2018). Low levels of muscarinic M1 receptor–positive neurons in cortical layers III and V in Brodmann areas 9 and 17 from individuals with schizophrenia. *Journal of Psychiatry and Neuroscience*, *43*(5), 338–346. https://doi.org/10.1503/jpn.170202

Schöbi, D., Homberg, F., Frässle, S., Endepols, H., Moran, R. J., Friston, K. J., Tittgemeyer, M., Heinzle, J., and Stephan, K. E. (2021). Model-based prediction of muscarinic receptor function from auditory mismatch negativity responses. *NeuroImage*, *237*(May). https://doi.org/10.1016/j.neuroimage.2021.118096

Shekhar, A., Potter, W. Z., Lightfoot, J., Lienemann, J., Dubé, S., Mallinckrodt, C., Bymaster, F. P., McKinzie, D. L., and Felder, C. C. (2008). Selective Muscarinic Receptor Agonist Xanomeline as a Novel Treatment Approach for Schizophrenia. *American Journal of Psychiatry*, *165*(8), 1033–1039. https://doi.org/10.1176/appi.ajp.2008.06091591

Shirazi-Southall, S., Rodriguez, D. E., and Nomikos, G. G. (2002). Effects of Typical and Atypical Antipsychotics and Receptor Selective Compounds on Acetylcholine Efflux in the Hippocampus of the Rat. *Neuropsychopharmacology*, *26*(5), 583–594. https://doi.org/10.1016/S0893-133X(01)00400-6

Stefanics, G., Heinzle, J., Horváth, A. A., and Stephan, K. E. (2018). Visual Mismatch and Predictive Coding: A Computational Single-Trial ERP Study. The Journal of Neuroscience : The Official Journal of the Society for Neuroscience, 38(16), 4020–4030. https://doi.org/10.1523/JNEUROSCI.3365-17.2018

Stephan, K. E., Baldeweg, T., and Friston, K. J. (2006). Synaptic Plasticity and Dysconnection in Schizophrenia. *Biological Psychiatry*, *59*(10), 929–939. https://doi.org/10.1016/j.biopsych.2005.10.005

Stephan, K. E., Friston, K. J., and Frith, C. D. (2009). Dysconnection in Schizophrenia: From abnormal synaptic plasticity to failures of self-monitoring. *Schizophrenia Bulletin*, *35*(3), 509–527. https://doi.org/10.1093/schbul/sbn176

Todd, J., Heathcote, A., Whitson, L. R., Mullens, D., Provost, A., and Winkler, I. (2014). Mismatch negativity (MMN) to pitch change is susceptible to order-dependent bias. *Frontiers in Neuroscience*, *8*, 180. https://doi.org/10.3389/fnins.2014.00180

Tzavara, E. T., Bymaster, F. P., and Nomikos, G. G. (2006). The procholinergic effects of the atypical antipsychotic olanzapine are independent of muscarinic autoreceptor inhibition. *Molecular Psychiatry*, *11*(7), 619–621. https://doi.org/10.1038/sj.mp.4001834

Wang, J., Chen, T., Jiao, X., Liu, K., Tong, S., and Sun, J. (2021). Test-retest reliability of duration-related and frequency-related mismatch negativity. *Neurophysiologie Clinique*, *51*(6), 541–548. https://doi.org/10.1016/j.neucli.2021.10.004

Weber, L. A., Diaconescu, A. O., Mathys, C., Schmidt, A., Kometer, M., Vollenweider, F., and Stephan, K. E. (2020). Ketamine Affects Prediction Errors about Statistical Regularities: A Computational Single-Trial Analysis of the Mismatch Negativity. The Journal of Neuroscience : The Official Journal of the Society for Neuroscience, 40(29), 5658–5668. https://doi.org/10.1523/JNEUROSCI.3069-19.2020

Weiner, D. M., Meltzer, H. Y., Veinbergs, I., Donohue, E. M., Spalding, T. A., Smith, T. T., Mohell, N., Harvey, S. C., Lameh, J., Nash, N., Vanover, K. E., Olsson, R., Jayathilake, K., Lee, M., Levey, A. I., Hacksell, U., Burstein, E. S., Davis, R. E., and Brann, M. R. (2004). The role of M1 muscarinic receptor agonism of N-desmethylclozapine in the unique clinical effects of clozapine. *Psychopharmacology*, *177*(1), 207–216. https://doi.org/10.1007/s00213-004-1940-5

Yohn, S. E., and Conn, P. J. (2018). Positive allosteric modulation of M1 and M4 muscarinic receptors as potential therapeutic treatments for schizophrenia. *Neuropharmacology*, *136*, 438–448. https://doi.org/10.1016/j.neuropharm.2017.09.012